# ELICIT: LLM AUGMENTATION VIA EXTERNAL IN-CONTEXT CAPABILITY

**Futing Wang** [1,2] * **Jianhao Yan** [1,2] * **Yue Zhang** [2,3] † **Tao Lin** [2,4] †
Zhejiang University [1]   Westlake University [2]
Institute of Advanced Technology, Westlake Institute for Advanced Study [3]
Research Center for Industries of the Future, Westlake University [4]
{wangfuting, yanjianhao, zhangyue, lintao}@westlake.edu.cn

## ABSTRACT

Enhancing the adaptive capabilities of large language models is a critical pursuit in both research and application. Traditional fine-tuning methods require substantial data and computational resources, especially for enhancing specific capabilities, while in-context learning is limited by the need for appropriate demonstrations and efficient token usage. Inspired by the expression of in-context learned capabilities through task vectors and the concept of modularization, we propose ELICIT, a framework consisting of two modules designed to effectively store and reuse task vectors to elicit the diverse capabilities of models without additional training or inference tokens. Our comprehensive experiments and analysis demonstrate that our pipeline is highly transferable across different input formats, tasks, and model architectures. ELICIT serves as a plug-and-play performance booster to enable adaptive elicitation of model capabilities. By externally storing and reusing vectors that represent in-context learned capabilities, ELICIT not only demonstrates the potential to operate modular capabilities but also significantly enhances the performance, versatility, adaptability, and scalability of large language models. Our code is publicly available [1].

## 1 INTRODUCTION

Large Language Models (LLMs) have revolutionized the field of Natural Language Processing (NLP), demonstrating remarkable versatility in tackling a wide array of tasks and real-world challenges (Devlin, 2018; Brown, 2020; Han et al., 2021; Achiam et al., 2023; Touvron et al., 2023). The power of these models lies in their ability to seamlessly integrate various *capabilities*, from logical reasoning (Bommasani et al., 2021) to common sense understanding (Talmor et al., 2018). In our rapidly evolving world, a crucial aspect of LLM is the ability to efficiently adapt to new tasks or scenarios.

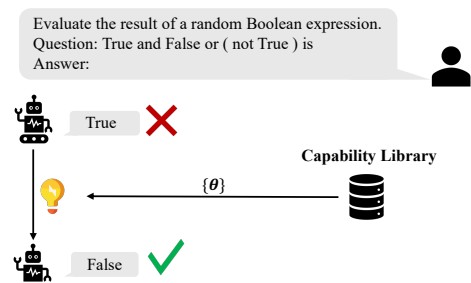

Figure 1: **Illustration of ELICIT**, which dynamically retrieves and integrates task vectors from a capability library to augment a language model's performance on arbitrary queries, without increasing token usage during inference.

Traditional fine-tuning methods, while effective in enhancing specific model capabilities (Devlin, 2018; Thirunavukarasu et al., 2023; Gururangan et al., 2020), often fall short in providing the necessary adaptability. These approaches are computationally intensive, leaving LLMs ill-equipped to handle the dynamic nature of real-world applications. In-Context Learning (ICL) (Brown, 2020) has emerged as a promising alternative, allowing LLMs to adapt to new tasks without additional training by leveraging their inherent capabilities (Team et al., 2023; Vacareanu et al., 2024; Agarwal

---

* These authors contributed equally to this work.
† Corresponding author.
[1] https://github.com/LINs-lab/ELICIT

et al., 2024). ICL, while effective, relies on carefully crafted demonstrations, requires extra overhead for each inference, and interrupts the context, potentially limiting its efficiency and flexibility (Lu et al., 2021; Zhao et al., 2021; Dong et al., 2022; Liu et al., 2023b).

We envision a next step in LLM adaptation: *Can we elicit and harness the potential of LLMs' inherent capabilities when adapting to new tasks, as demonstrated by ICL, while simultaneously maintaining efficiency and flexibility?*

Our research explores this question by introducing a novel approach ELICIT inspired by the concept of modularization (Pfeiffer et al., 2023; Fedus et al., 2022; Ding et al., 2023; Zhang et al., 2023b; Xiao et al., 2024). ELICIT involves the establishment of a comprehensive library of task vectors, each eliciting one in-context capability within the LLM. For any arbitrary query text, ELICIT dynamically leverage this capability library to selectively elicit the capability, effectively and flexibly tapping into the model's inherent capabilities on an as-needed basis. We empirically verify the advantanges of ELICIT under 20 tasks and 4 models:

- **Efficient Capability Elicitation**: ELICIT aims to improve the model's task-specific capabilities with minimal additional computational cost during inference. Across 20 tasks and 4 models, ELICIT achieves an average improvement of 11.4% over zero-shot performance while maintaining the same token usage (Section 4.2).
- **Flexible Task Handling**: ELICIT can adapt to various tasks without requiring task-specific demonstrations or rigid templates, enhancing performance on both in-domain and unseen tasks (Sections 4.2 and 4.4).
- **Selective Capability Activation**: ELICIT allows for targeted activation of specific model capabilities based on the input query. In our experiments with a math-only capability library, ELICIT boosted Math performance dramatically (e.g., from 2.6% to 21.3% for Mistral) while maintaining or slightly improving performance in other domains (Section 4.3).
- **Complementary Integration**: ELICIT shows potential for complementary use with existing methods, offering further performance gains. When combined with BM25 retrieval, ELICIT enhanced Pythia's average performance from 22.1% to 28.3% (Section 4.5).

**Our key contributions are summarized as follows:**

- We introduce a novel, modular framework for enhancing the adaptive capabilities of LLMs on demand with minimal computational overhead.
- We conduct extensive experiments to evaluate our method, showcasing its effectiveness across different query formats, language models, and tasks.
- We provide a thorough analysis of our method, offering insights into the design choices and their contributions to overall performance.

## 2 RELATED WORK

**In-Context Learning.** While Brown (2020) introduced In-Context Learning (ICL) as a simple yet effective way to enhance LLM performance by incorporating demonstrations into prompts, its applications have rapidly expanded across diverse domains. ICL enables model to adapt to a wide array of tasks ranging from traditional NLP benchmarks to more specialized tasks such as egression (Vacareanu et al., 2024), kNN classification (Agarwal et al., 2024; Dinh et al., 2022), and even jailbreaking (Anil et al., 2024). Researchers have actively explored various avenues to further enhance ICL's adaptability and effectiveness. These efforts include increasing demonstration quantity (Bertsch et al., 2024; Agarwal et al., 2024; Zhang et al., 2023a; Team et al., 2023), fine-tuning models for ICL (Min et al., 2021), leveraging prompt engineering (Nie et al., 2022), and implennmenting demonstration retrieval (Liu et al., 2021; Rubin et al., 2021; Li et al., 2023; Shi et al., 2022).

Concurrently, deeper insights into ICL's underlying mechanisms have been sought through diverse perspectives. Some researchers view ICL as a process of compressing training data into task-specific vectors (Hendel et al., 2023), while others relate it to gradient descent (Von Oswald et al., 2023) or analyze it through the lens of repetition (Yan et al., 2023) and memorization (Golchin et al., 2024). Building upon ICL advancements, we explored eliciting and harnessing LLMs' inherent capabilities for new task adaptation, akin to ICL, while maintaining efficiency and flexibility.

**Task representation for ICL.** Inspired by findings that intermediate representations in LLMs encode semantic meaning (Zou et al., 2023), researchers have explored injecting in-context learning

demonstrations, encoded as function vectors, into intermediate representations to trigger desired predictions (Liu et al., 2023b; Hendel et al., 2023; Todd et al., 2023; Li et al., 2024). The scope of this research line has broadened to include different modalities,with recent work demonstrating its effectiveness in both visual (Hojel et al., 2025) and multimodal domains (Huang et al., 2024). However, this line of work focuses on manipulating internal representations. We are the first to comprehensively explore the modular approach of externally storing and retrieving such task representations to augment large language model capabilities.

**Modular LLM.** Examining and understanding the modular nature of large language models (LLMs) has become a crucial area of study for researchers (Pfeiffer et al., 2023; Fedus et al., 2022; Ding et al., 2023; Zhang et al., 2023b; Xiao et al., 2024). Initial investigations suggest that LLMs possess the capability to be broken down into distinct specialized components or modules. Some approaches introduce additional modules or parameters for optimization, including parameter-efficient tuning techniques like adapter layers (Houlsby et al., 2019; Pfeiffer et al., 2020), prompt tuning (Liu et al., 2023a; Ding et al., 2021), and parameter subset optimization methods such as BitFit (Zaken et al., 2021) and binary masks (Guo et al., 2020; Zhao et al., 2022). Other approaches involve training dedicated models for task composition (Shao et al., 2023; Mu et al., 2024) or merging fine-tuned parameter adjustments (Ilharco et al., 2022; Panigrahi et al., 2023; Merullo et al., 2023; Yu et al., 2024). Inspired by such modular perspectives, we explore the question of using task vectors in a modular way to dynamically elicit capabilities within the model.

## 3 METHOD

To elicit the hidden capability inside LLMs, we build our ELICIT by introducing a capability library which condenses each in-context learned capability into a task vector, and utilizing a retrieval module to strengthen the model when a task vector is helpful.

This section describes our implementation of ELICIT. We first formally define in-context learning Task Vectors (Section 3.1), and motivate our work. Then, we discuss the design choices of building capabilities libraries (Section 3.2), including the layer selection and intervention strategies. Finally, we introduce our retrieval module (Section 3.3) to dynamic elicit and leverage model's capability.

### 3.1 FROM ICL TO TASK VECTORS: FORMAL DEFINITIONS

**In-Context Learning (ICL).** Firstly, we define the framework for ICL. Let $\mathcal{T}$ represent a collection of tasks. For each task $t \in \mathcal{T}$, there exists a dataset $\mathbf{P}_t$ of in-context prompts. Each prompt $\mathbf{p}_i^t \in \mathbf{P}_t$ is a sequence of tokens that represents the $i$-th prompt for task $t$. Specifically, each prompt $\mathbf{p}_i^t$ consists of two components: (1) a set of $N$ input-output demonstrations $\mathbf{D} = (\mathbf{x}_{ij}, \mathbf{y}_{ij})_{j=1}^N$ from task $t$, where $j$ indexes the sequence of pairs ranging from 1 to $N$, and (2) a query input $\mathbf{x}_{iq}$, which is distinct from the inputs in $\mathbf{D}$. We formally represent an ICL prompt $\mathbf{p}_i^t$ as:

$$\mathbf{p}_i^t = [(\mathbf{x}_{i1}, \mathbf{y}_{i1}), \ldots, (\mathbf{x}_{iN}, \mathbf{y}_{iN}), \mathbf{x}_{iq}]. \tag{1}$$

The Language Model (LM) aims to predict the corresponding target response $\mathbf{y}_{iq}$ for the query input $\mathbf{x}_{iq}$. Through learning from the demonstrated input-output mappings in $\mathbf{D}$, ICL can enhances the model's capability to perform this task. We firstly introduce the hidden state in Transformers below.

**Task Vector.** Previous research (Hendel et al., 2023) introduced the concept of a task vector in the context of ICL. We build upon this foundation in our work. We first introduce the definition of hidden state representations in transformer models and task vector is derived from it.

**Definition 3.1** (**Hidden State Representation in Transformers**). *Let $T$ be an auto-regressive transformer language model with $L$ layers. For each layer $l \in 1, \ldots, L$, we define $\mathbf{h}_l \in \mathbb{R}^d$ as the vector representation of the last token at layer $l$. The computation of $\mathbf{h}_l$ follows the recurrence relation (Vaswani, 2017): $\mathbf{h}_l = \mathbf{h}_{l-1} + \mathbf{m}_l + \mathbf{a}_l$, where $\mathbf{m}_l$ is the output of a multilayer perceptron at layer $l$, and $\mathbf{a}_l$ is the projection of the attention output into the hidden state at layer $l$.*

Having established the notion of hidden states in transformer models, we can now formally define the task vector within the ICL framework.

**Definition 3.2** (**Task Vector $\theta$**). *ICL functions by learning a task-specific mapping from demonstrations. This mapping is represented as a task vector $\theta$. The task vector is derived from the activation state $\mathbf{h}_l$ (as defined in Definition 3.1) at a specific layer $l$, corresponding to the last token of the prompt. This vector subsequently steers the transformer to yield pertinent outputs for given queries.*

The task vector, as defined, encapsulates the essence of the task. This leads to the following lemma, highlighting its role in simulating ICL behavior.

**Lemma 3.3** (**Task Vector for ICL Simulation**). *Given a task vector $\theta$ that effectively captures the information from demonstrations in an ICL setting, we can simulate the behavior of regular ICL with only query as follows:*

$$T[\mathbf{p}_i^t] \approx f(\theta; \mathbf{x}_{iq}),$$

*where:*

- $T[\mathbf{p}_i^t]$ *represents the output of the transformer model given a ICL prompt $\mathbf{p}_i^t$ defined as (1).*
- $f(\theta; \mathbf{x}_q)$ *denotes a function that processes the query input $\mathbf{x}_q$ in a zero-shot manner, guided by the information encoded in the task vector $\theta$.*

**Remark 3.4** (**Intervention of Task Vector $\theta$**). *The function $f(\theta; \mathbf{x}_q)$ mentioned in Lemma 3.3 is an abstract concept expressing that the task vector can be used to influence the model's inference process. In practice, $f(\theta; \mathbf{x}_q)$ is implemented through operations on the hidden states $\mathbf{h}_l$ and the task vector $\theta$. Specifically, these operations can take the following forms:*

1. ***Replacement*** *(Hendel et al., 2023): The task vector $\theta$ directly replaces the hidden state $\mathbf{h}_l$, i.e., $\tilde{\mathbf{h}}_l = \theta$.*
2. ***Linear combination*** *(Todd et al., 2023): The task vector $\theta$ is combined linearly with the hidden state $\mathbf{h}_l$, i.e., $\tilde{\mathbf{h}}_l = \mathbf{h}_l + \alpha\theta$, where $\alpha$ is an adjustable scalar parameter.*

While previous research has demonstrated the existence and extractability of task vectors, it also has shown the potential for serving a technique to elicit the inherent capabilities when adapting to difference tasks as ICL, while simultaneously maintaing computational efficiency and flexibility.

We investigate the research problem through task vectors by proposing ELICIT, a framework designed to leverage these vectors for enhancing model capabilities. As shown in Figure 2, ELICIT consists of two main components:

- **Build Capability Library**: A capability library that stores task vectors representing various in-context learned capabilities.
- **Dynamic Capability Elicitation**: A dynamic retrieval module that selectively activates relevant task vectors based on the input query.

## 3.2 BUILDING CAPABILITIES LIBRARY

To investigate the idea of ELICIT, we first create a library of in-context learned capabilities $\Theta = \{\{\theta_i^t\}_{i=1}^k\}_{t \in \mathcal{T}}$. Each element in this library is represented by a task vector (as defined in Definition 3.2). Here, $k$ denotes the number of ICL prompts for each task $t$, and we use $k = 10$ for illustration. Definition 3.2 describes $\theta \in \mathbb{R}^d$ as a single layer's hidden state. In our implementation, we collect $\theta \in \mathbb{R}^{L \times d}$, which includes representations from all $L$ layers, to enable the exploration of various designs for the sequential components of ELICIT.

The implementation of creating capability library involves two critical considerations: ① *Dynamic Layer Selection for $l^*$*, and ② *Intervention Strategies*. ① determines the appropriate layer $l$ to intervene into during further reuse, utilizing the corresponding task vector, while ② decides how to appropriately intervene the task vector to influence the model's behavior (possible methods are described in Remark 3.4). Our framework addresses these considerations as follows.

① **Dynamic Layer Selection for $l^*$.** The selection of the optimal layer for task vector intervention is crucial for maximizing the effectiveness of our approach (Todd et al., 2023; Hendel et al., 2023). Appendix B further illustrates the variation in the optimal layer across different tasks. We implement a dynamic layer selection strategy to determine the optimal layer $l^*$ for task vector intervention. While using a validation set to identify the optimal layer is not a novel concept, *our contribution lies in addressing the challenge of determining the intervention layer when applying the library in our*

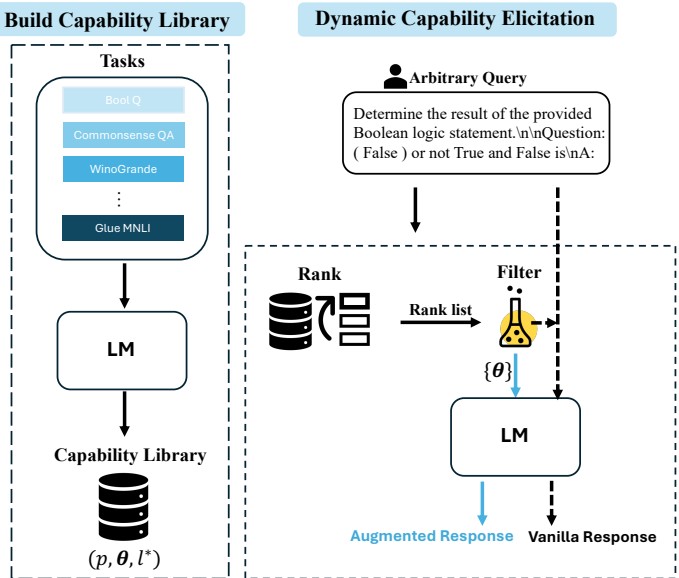

Figure 2: **Overview of the proposed ELICIT framework for Large Language Model Augmentation.**
ELICIT consists of two modular components: (1) Build Capability Library - constructing a library of task-specific task vectors by learning from diverse task; (2) Dynamic Capability Elicitation - dynamically retrieving and integrating relevant task vectors from the library to augment the model's capability for an arbitrary input query.

*proposed pipeline: we equip each task vector with its corresponding optimal layer, pre-identified during the library construction phase, thereby enabling efficient and effective reuse of task vectors during inference.* Our process is as follows:

- We obtain the task vector $\boldsymbol{\theta} \in \mathbb{R}^{L \times d}$ by processing the ICL prompt $\mathbf{p}_i^t$ (defined in (1), using randomly selected $N = 16$ demonstrations). This is done by extracting the hidden states $\{\mathbf{h}_l\}_{l=1}^{L}$ from all layers of the language model. We also store these ICL prompts for future use.
- Using the validation set, we perform a layer-by-layer analysis:
  - For each layer $l$, we intervene $\boldsymbol{\theta}_l$ during zero-shot query processing.
  - We measure zero-shot accuracy on the validation set for each intervention.
  - We identify $l^*$ as the layer yielding the best accuracy.
- We repeat this process for each task-model combination, creating a library where each task vector $\boldsymbol{\theta}$ has its corresponding best layer $l^*$.

When reusing task vectors in library, for any $\boldsymbol{\theta} \in \boldsymbol{\Theta}$, we intervene at the pre-identified optimal layer $l^*$ for each task-model combination. This dynamic selection method ensures the performance of the task vector and provides a generalizable framework adaptable to different tasks.

②  **Intervention Strategies.** The concept of intervention, formally introduced in Remark 3.4, also outlines two methods for incorporating the task vector $\boldsymbol{\theta}_l$ into the model's inference process. We evaluate these two intervention strategies: 1) linear combination of the original hidden state and task vector with varying intervene strength $\alpha$, and 2) direct replacement of the original hidden state with the task vector. We examine the impact of these intervention strategies on both task performance and language modeling capability, with the latter measured using cross-entropy loss on the pre-training dataset (i.e, WikiText).

Figure 3 provides a detailed visualization of how varying $\alpha$ affect both accuracy and cross-entropy loss in the Llama3-8B model across a diverse set of 20 tasks. *Results reveal a clear trade-off between task performance and language modeling capability as intervention strength increases.* Among the strategies tested, the additive approach $\tilde{\mathbf{h}}_l = \mathbf{h}_l + 2 \times \boldsymbol{\theta}_l$ consistently demonstrates superior performance across a wide range of tasks while minimizing degradation in language modeling ability. Results for other models are presented in the Appendix A, showing similar trends.

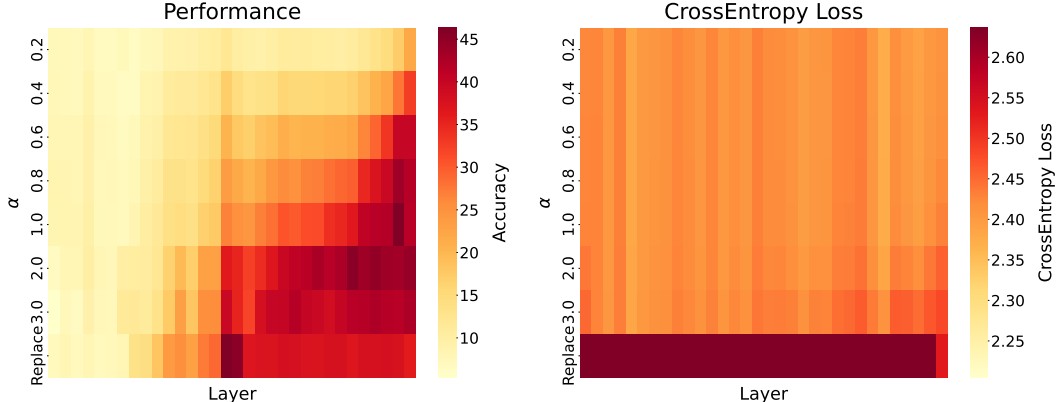

Figure 3: **Varying intervention strengths affect accuracy and cross-entropy loss in Llama3-8B on valid set of** 20 **tasks across different layer**. Higher in intervention strengths improve average task performance across layers but negatively impact language modeling capabilities. This reveals a trade-off between task-specific enhancement and general language modeling proficiency using task vectors.

**In conclusion.** Our library contains $k \times |\mathcal{T}|$ items for each model, each consisting of three key components: (1) the ICL prompt $\mathbf{p}_i^t$, (2) the corresponding task vector $\boldsymbol{\theta} \in \mathbb{R}^{L \times d}$, and (3) the pre-identified optimal layer $l^*$.

### 3.3 DYNAMIC CAPABILITY ELICITATION

After the creation of the capability library, as described in Section 3.2, we consider two considerations: ⓘ *Relevant Task Vector Selection*, and ⓘⓘ *Threshold-Based Filtering*. ⓘ Relevant Task Vector Selection focuses on identifying the most relevant task vectors from the library for a given test query $q$. We aim to find the most relevant task vectors $\boldsymbol{\theta}^q \in \boldsymbol{\Theta}$ stored in the library. Unlike traditional in-context learning (ICL), we lack meta-information about the query. ⓘⓘ Threshold-Based Filtering determines whether to utilize a retrieved task vector or not, to avoid compromising performance when no suitable task vectors are available in the library. Our framework addresses these challenges as follows:

ⓘ **Relevant Task Vector Selection.** We address the challenge of selecting the most relevant task vectors by employing a binary classifier to calculate similarity scores. This classifier is built upon the SimCSE RoBERTa model[2], augmented with a 2-layer Multi-Layer Perceptron (MLP) head. The architecture incorporates ReLU activation functions and a dropout rate of $0.2$ for regularization.

We fine-tuned this model over 15 epochs using a learning rate of $2e{-}5$ on our curated dataset of 10,000 examples. The trained classifier is then used to compute similarity scores between a given query and each ICL prompt in our library. These scores are used to rank all library items, producing a similarity list of size $k \times |\mathcal{T}|$. The top-ranked task vector from this list is selected as our target for further processing.

ⓘⓘ **Threshold-Based Filtering.** To determine whether to utilize stored task vectors from our library, we implement a threshold-based approach using similarity scores. This threshold is established through a comprehensive analysis of the recall-precision trade-off across our validation set, as illustrated in Figures 4a, utilizing the aggregated similarity lists for all samples. The AUC scores of our precision-recall curves (i.e., $0.96$) demonstrate the high effectiveness of our threshold-based approach in accurately determining whether stored task vectors require intervention. Our evaluation of various recall levels, as shown in Figures 4b, reveals that a recall of $0.8$ provides the optimal balance for our pipeline, other models' results shown in Appendix D.

Our decision process of how to utilize the similarity list and threshold to choose whether to use the stored task vectors and what task vectors to apply is as follows:

- We implement Dynamic Top-K Thresholding (DTT). If the highest similarity score exceeds the threshold, we select the top 10 task vectors from the ranked list for further processing.

---

[2]princeton-nlp/sup-simcse-roberta-base

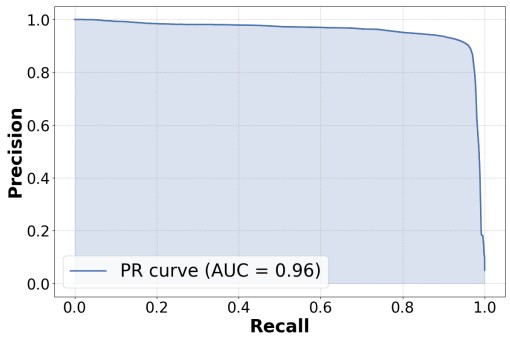 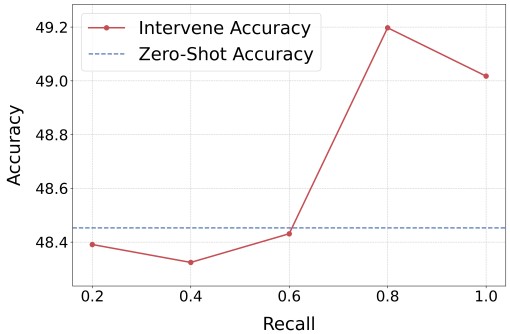

(a) Precision-Recall Curves                    (b) Valid Performance varying different recall.

Figure 4: **Precision-Recall Curves and recall sweeping on Llama3-8B in valid set across 20 tasks.** (a) Precision-Recall curves for the retriever across 20 tasks (AUC=0.96), guiding threshold selection for high recall and precision. (b) Validation set accuracy after intervention using different recall thresholds.

Table 1: **Models used in this work.** We consider decoder-only auto-regressive language models and recurrent neural networks that are capable of ICL. For each model, we present the number of parameters, context window during training, and the number of layers $|L|$.

| Model | HuggingFace ID | Citation | Parameters | Train Length | $|L|$ |
|---|---|---|---|---|---|
| Llama 3 | meta-llama/Meta-Llama-3-8B | Dubey et al. (2024) | 8B | 8k | 32 |
| Mistral | TIGER-Lab/Mistral-7B-Base-V0.2 | Jiang et al. (2023) | 7B | 32k | 32 |
| Pythia | EleutherAI/pythia-2.8b | Biderman et al. (2023) | 2.8B | 2k | 32 |
| Mamba | state-spaces/mamba-2.8b-hf | Gu & Dao (2023) | 2.8B | 2k | 64 |

- We then employ a majority voting mechanism among the optimal layers suggested by these top vectors to determine the final layer for intervention.
- In cases where the highest similarity score falls below the threshold, we refrain from using any stored task vector, relying instead on the model's base capabilities.

## 4 EXPERIMENTS

To comprehensively evaluate the effectiveness of ELICIT, we conduct a series of experiments designed to explore the following key questions:

- *Capability Elicitation Efficiency:* Can ELICIT effectively elicit the model's capabilities without incurring significant additional computational costs?
- *Selective Activation:* Is ELICIT capable of selectively activating relevant capabilities as needed for specific tasks?
- *Complementarity:* How well does ELICIT integrate with and complement existing methods in the field?
- *Generalization:* Can ELICIT handle novel queries, particularly those that diverge significantly from the task vectors currently stored in the library?

### 4.1 EXPERIMENT SETUP

**Model.** We utilize decoder-only auto-regressive language models (Pythia-2.8B (Biderman et al., 2023), LLaMA3-8B (Dubey et al., 2024), and Mistral-7B (Jiang et al., 2023)) and recurrent neural network (Mamba-2.8B (Gu & Dao, 2023)). Table 1 provides a comprehensive overview of these models, detailing their key characteristics including the number of parameters, layer numbers, and training context window size. For all models, we use the corresponding `huggingface` implementations (Wolf et al., 2020).

**Tasks.** To assess the efficacy of our proposed pipeline across a diverse array of scenarios, we have meticulously constructed a benchmark comprising 20 distinct tasks. This benchmark is designed to evaluate the model's performance on both classification and multiple-choice problems, spanning a wide spectrum of applications and complexities. The tasks are categorized into five domains:

- **Knowledge**: CommonsenseQA (Talmor et al., 2018), OpenBookQA (Mihaylov et al., 2018), HellaSwag (Zellers et al., 2019), and BoolQ (Clark et al., 2019);

Table 2: **Performance of ELICIT across model and tasks.** ELICIT significantly enhances performance while maintaining the same token usage as Zero-shot, often achieving results comparable to or better than 16-shot and 16-shot BM25 ICL retriever methods. This improvement is consistent across various models and tasks, demonstrating ELICIT's efficiency and effectiveness in boosting model capabilities without increasing computational demands. We sample 100 examples per task across three random seeds.

| Model | | # Tokens | NLU | Reasoning | Knowledge | Math | Safety | Avg. |
|---|---|---|---|---|---|---|---|---|
| **Llama3** | 16-shot | 1883.8 ± 0.9 | 60.6 ± 1.0 | 56.0 ± 0.4 | 70.6 ± 1.0 | 26.7 ± 2.0 | 62.1 ± 0.4 | 55.2 ± 0.4 |
| | bm25 | 2350.7 ± 24.9 | 56.1 ± 1.5 | 68.8 ± 0.2 | 69.5 ± 0.9 | 28.0 ± 2.3 | 56.7 ± 2.0 | 55.8 ± 0.7 |
| | Zero-shot | 108.3 ± 1.4 | 32.2 ± 1.2 | 31.6 ± 0.2 | 42.5 ± 1.2 | 14.0 ± 1.0 | 35.5 ± 1.2 | 31.2 ± 0.7 |
| | ELICIT | 108.3 ± 1.4 | **41.6 ± 0.4** | **46.7 ± 0.1** | **60.6 ± 1.4** | **19.1 ± 1.4** | **49.9 ± 2.1** | **43.5 ± 0.8** |
| **Mistral** | 16-shot | 2161.3 ± 0.9 | 55.3 ± 0.5 | 52.1 ± 0.5 | 70.8 ± 0.4 | 23.7 ± 1.7 | 63.1 ± 0.6 | 53.0 ± 0.1 |
| | bm25 | 2655.2 ± 27.3 | 55.2 ± 0.3 | 66.0 ± 0.5 | 70.2 ± 1.9 | 24.1 ± 0.4 | 62.1 ± 0.5 | 55.5 ± 0.4 |
| | Zero-shot | 123.5 ± 1.7 | 29.6 ± 1.2 | 26.9 ± 0.4 | 45.5 ± 1.3 | 2.8 ± 0.1 | 36.1 ± 0.3 | 28.2 ± 0.5 |
| | ELICIT | 123.5 ± 1.7 | **41.9 ± 1.0** | **48.3 ± 0.3** | **59.4 ± 0.9** | **20.3 ± 0.9** | **48.7 ± 1.8** | **43.7 ± 0.6** |
| **Pythia** | 16-shot | 1942.4 ± 0.9 | 50.2 ± 0.5 | 19.6 ± 0.1 | 12.8 ± 0.9 | 9.2 ± 1.6 | 31.8 ± 0.9 | 24.7 ± 0.2 |
| | bm25 | 2422.8 ± 26.0 | 33.3 ± 2.2 | 25.8 ± 0.4 | 12.9 ± 0.5 | 11.0 ± 1.8 | 27.3 ± 2.1 | **22.1 ± 0.5** |
| | Zero-shot | 110.0 ± 1.5 | 43.0 ± 0.4 | 18.3 ± 0.3 | 22.0 ± 1.5 | 7.3 ± 0.1 | 32.5 ± 1.2 | 24.6 ± 0.4 |
| | ELICIT | 110.0 ± 1.5 | **64.0 ± 1.6** | **23.6 ± 1.1** | **20.4 ± 1.4** | **14.5 ± 1.0** | **41.2 ± 2.5** | **32.7 ± 0.5** |
| **Mamba** | 16-shot | 1942.4 ± 0.9 | 37.5 ± 1.0 | 31.5 ± 0.5 | 31.6 ± 0.8 | 14.2 ± 0.5 | 41.7 ± 1.2 | 31.3 ± 0.3 |
| | bm25 | 2422.8 ± 26.0 | 29.3 ± 2.2 | 34.9 ± 0.9 | 24.7 ± 0.5 | 15.1 ± 2.2 | 35.4 ± 1.2 | 27.9 ± 0.3 |
| | Zero-shot | 110.0 ± 1.5 | 36.1 ± 1.5 | 19.3 ± 0.5 | 17.3 ± 1.2 | 5.8 ± 1.2 | 30.1 ± 0.1 | 21.7 ± 0.2 |
| | ELICIT | 110.0 ± 1.5 | **51.1 ± 0.7** | **28.7 ± 0.8** | **29.2 ± 1.3** | **15.3 ± 1.1** | **48.2 ± 1.8** | **34.5 ± 0.6** |

- **Reasoning**: Four subsets from Big-Bench Hard (BBH) (Suzgun et al., 2022) and ARC-Challenge (Clark et al., 2018);
- **Mathematics**: MathQA (Amini et al., 2019) and MMLU Pro-MATH (Wang et al., 2024);
- **Safety**: Crows-Pairs (Nangia et al., 2020), BBQ-Age (Parrish et al., 2021), Ethics-Commonsense, and Ethics-Justice (Merity et al., 2016);
- **Natural Language Understanding (NLU)**: GLUE (SST-2, QNLI, MNLI) (Wang, 2018) and SuperGLUE (WIC, RTE) (Wang et al., 2019).

**Evaluation.** To evaluate ELICIT under a real usage scenario, where the demonstrations can hardly be at the same format with the test query, we augment the test query with two additional formats different from the demonstration in library. Furthermore, in our preliminary experiments, we find that zero-shot LLMs cannot answer properly with contextual guidance. Thus, to ensure a fair comparison with the zero-shot scenario, we add task templates before the test query. More details and examples can be found in Appendix E.

**Baselines.** Our primary baseline is the zero-shot performance of LLMs, as our method maintains the same token usage. For reference, we also include in-context learning (ICL) and BM25 (Robertson et al., 2009) retrieval of 16 examples from the same pool of examples used in constructing the capability library. However, these are not directly comparable to our method, due to the raised nearly 20 times more tokens consuming. The ICL baseline is task-specific, requiring knowledge of each query's task type to use corresponding demonstrations. In contrast, our method is task-agnostic, applicable across various tasks without needing task-specific information or prompts.

## 4.2 EFFICIENT CAPABILITY ELICITATION

**ELICIT *achieves efficiently eliciting models' capabilities.*** From Tables 2, comparing the zero-shot baseline and ELICIT, we observe that ELICIT significantly elicits model capabilities across most tasks without increasing token usage. Across the 20 tasks, ELICIT achieves an average improvement of 11.4% across different models. For Llama3, ELICIT improves over zero-shot by 12.3% while using the same 108.2 tokens. ELICIT demonstrates substantial gains in Reasoning (e.g., +15.1% for LLama3) and Safety tasks (e.g., +14.4% for LLama3). In some cases, ELICIT's performance is comparable to or surpasses that of 16-shot and BM25 methods, despite their higher token requirements. Furthermore, it exhibits robustness across various template formats, highlighting its versatility.

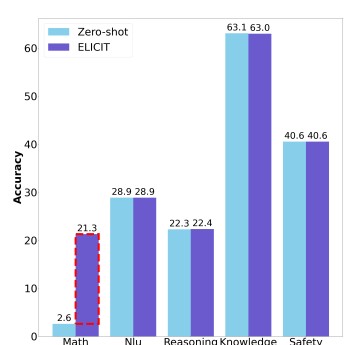

Figure 5: **Performance on ELICIT across different domains** when the library only contains math-related task vectors on Mistral.

Table 3: **ELICIT can generalize to unseen tasks.** ELICIT achieves significant performance gains without additional token usage across different models and unseen tasks. We sample 100 examples per task across three random seeds. We use BM25 retrieval of 16 examples as baseline.

| | | # Tokens | GLUE COLA | BBQ Religion | Deepmind | MMLU-Psychology | BBH-five-objects | Avg |
|---|---|---|---|---|---|---|---|---|
| **Llama** | BM25 | 2502.8 ± 26.0 | 55.4 ± 1.0 | 64.6 ± 1.3 | 30.7 ± 1.7 | 83.0 ± 0.1 | 48.3 ± 0.0 | 56.4 ± 0.4 |
| | Zero-shot | 103.6 ± 47.7 | **72.0 ± 0.7** | 38.6 ± 1.1 | 17.5 ± 2.6 | 54.2 ± 0.3 | 17.1 ± 0.0 | 39.9 ± 0.8 |
| | ELICIT | 103.6 ± 47.7 | 63.4 ± 0.9 | **45.0 ± 0.7** | **23.7 ± 3.4** | **70.0 ± 0.6** | **25.7 ± 0.0** | **45.6 ± 0.4** |
| **Mistral** | BM25 | 2804.6 ± 27.6 | 44.4 ± 2.2 | 70.7 ± 0.7 | 26.6 ± 3.9 | 78.7 ± 1.1 | 25.7 ± 0.0 | 49.2 ± 0.3 |
| | Zero-shot | 115.4 ± 51.0 | **43.3 ± 1.1** | 35.4 ± 3.3 | 9.0 ± 0.4 | 57.9 ± 0.7 | 7.4 ± 0.0 | 30.6 ± 1.0 |
| | ELICIT | 115.4 ± 51.0 | 41.7 ± 0.8 | **42.1 ± 2.5** | **25.1 ± 1.2** | **65.6 ± 0.6** | **15.6 ± 0.0** | **38.0 ± 0.6** |
| **Pythia** | BM25 | 2600.0 ± 28.3 | 5.8 ± 1.0 | 19.1 ± 1.2 | 14.1 ± 1.2 | 4.7 ± 0.3 | 1.0 ± 0.0 | 8.9 ± 0.3 |
| | Zero-shot | 106.7 ± 49.6 | **48.5 ± 0.6** | 21.7 ± 1.7 | 9.7 ± 1.2 | 20.1 ± 0.8 | 7.6 ± 0.0 | 21.5 ± 0.1 |
| | ELICIT | 106.7 ± 49.6 | 45.4 ± 0.6 | **30.3 ± 4.2** | **14.2 ± 1.8** | **20.4 ± 0.6** | **14.3 ± 0.0** | **24.9 ± 0.6** |
| **Mamba** | BM25 | 2600.0 ± 28.3 | 48.1 ± 3.1 | 30.6 ± 1.1 | 21.6 ± 3.3 | 19.1 ± 0.9 | 25.8 ± 0.0 | 29.0 ± 0.9 |
| | Zero-shot | 106.7 ± 49.6 | **70.3 ± 1.0** | 21.3 ± 2.9 | 10.9 ± 0.7 | 13.9 ± 0.5 | 6.2 ± 0.0 | 24.5 ± 0.4 |
| | ELICIT | 106.7 ± 49.6 | 63.6 ± 0.4 | **31.5 ± 2.5** | **22.1 ± 3.3** | **20.4 ± 0.2** | **14.4 ± 0.0** | **30.4 ± 0.9** |

## 4.3 Selective Adaptive Activation of Capabilities

**ELICIT *elicits capability when necessary.*** We demonstrate selective activation by constructing a library containing only math-related task vectors, as shown in Figure 5. The results clearly illustrate that ELICIT significantly boosts performance in the Math domain, with a dramatic increase from 2.6% to 21.3%, while maintaining performance in other domains for Mistral. Notably, the Reasoning domain also shows a slight improvement, increasing from 22.3% to 22.4%. This behavior stems from ELICIT's selective application of task vectors from library, which are not applied when no relevant tasks vectors are detected. More discussion is presented in Appendix J. Results for other models, presented in Appendix F, demonstrate a similar trend. The striking improvement in Math performance, coupled with the subtle gain in Reasoning and the stability in other domains, demonstrates ELICIT's capacity for targeted capability activation, making it a flexible and efficient performance enhancer.

## 4.4 ELICIT generalize to Unseen Tasks without addtional information

**ELICIT *generalizes across unseen tasks.*** In Table 3, we observe that ELICIT significantly improves model performance on unseen tasks (GLUE-COLA, BBQ Religion, Deepmind (Saxton et al., 2019), MMLU-Psychology, and BBH-Logical-Deduction-Five-objects) not present in its capability library. Across all models, ELICIT consistently outperforms the Zero-shot baseline. In several cases, it even approaches or surpasses the BM25 retrieval baseline, despite using substantially fewer tokens. For instance, on the MMLU-Psychology task, ELICIT achieves a 15.8% absolute improvement over Zero-shot for Llama3 Model. These results are achieved without additional token usage and task information, demonstrating ELICIT's efficiency, flexibility, and generalization ability.

## 4.5 Complementary Intergration

**ELICIT *shows potential as a plug-and-play performance booster.*** While ELICIT demonstrates compatibility with existing solutions like BM25 retrieval, Table 4 reveals nuanced performance patterns. For smaller models (Pythia-2.8B and Mamba-2.8B), combining ELICIT with BM25 yields consistent improvements, with Pythia's average performance increasing from 22.1% to 28.3% (+5.9%). However, larger models (Llama3-8B and Mistral-7B) exhibit mixed results: while NLU and Reasoning tasks show modest gains (e.g., +2.6% for Llama3), Knowledge and Safety tasks experience slight declines. Aligning with the findings of Li et al. (2024), this phenomenon can be attributed to two factors: (1) smaller models' relatively weak in-context learning capabilities benefit more from additional task-relevant information provided by our method, while (2) larger models' inherently stronger in-context adaptation abilities may be disrupted by the introduction of additional context that alters their learned representations. Future work could investigate this scale-dependent phenomenon.

# 5 Ablation Study

## 5.1 Similarity-Based Retrieve

We also explored similarity-based retrieval methods, such as cosine similarity, t-SNE distance, and Euclidean distance between the query embedding and the task vectors $\theta$ in capability library. However, as illustrated in Figure 6, the precision-recall curves for these methods on Llama3 exhibit

Table 4: **ELICIT as a potential plug-and-play performance booster**: performance when combined with BM25 on in-domain tasks. Results indicate stronger complementary effects for smaller models (Pythia, Mamba), while larger models (Llama3, Mistral) show task-specific variations.

| Model | | NLU | Reasoning | Knowledge | Math | Safety | Avg. |
|---|---|---|---|---|---|---|---|
| **Llama** | **BM25** | 56.1 ± 1.5 | **68.8 ± 0.2** | **69.5 ± 0.9** | **28.0 ± 2.3** | **56.7 ± 2.0** | **55.8 ± 0.7** |
| | **BM25+ELICIT** | **58.0 ± 0.4** | 62.7 ± 0.5 | 65.1 ± 0.5 | 25.1 ± 1.6 | 54.5 ± 3.5 | 53.1 ± 1.1 |
| **Mistral** | **BM25** | **55.2 ± 0.3** | **66.0 ± 0.5** | **70.2 ± 1.9** | 24.1 ± 0.4 | **62.1 ± 0.5** | **55.5 ± 0.4** |
| | **BM25+ELICIT** | 54.5 ± 0.8 | 62.6 ± 0.4 | 67.5 ± 1.7 | **24.8 ± 1.9** | 58.0 ± 1.4 | 53.5 ± 0.5 |
| **Pythia** | **BM25** | 33.3 ± 2.2 | 25.8 ± 0.4 | 12.9 ± 0.5 | 11.0 ± 1.8 | 27.3 ± 2.1 | 22.1 ± 0.5 |
| | **BM25+ELICIT** | **53.5 ± 1.5** | **26.5 ± 1.1** | **14.5 ± 0.6** | **13.2 ± 1.7** | **33.7 ± 0.7** | **28.3 ± 0.3** |
| **Mamba** | **BM25** | 29.3 ± 2.2 | **34.9 ± 0.9** | 24.7 ± 0.5 | 15.1 ± 2.2 | 35.4 ± 1.2 | 27.9 ± 0.3 |
| | **BM25+ELICIT** | **38.4 ± 1.2** | 31.6 ± 0.4 | **28.9 ± 0.2** | **15.2 ± 3.8** | **42.9 ± 1.9** | **31.4 ± 0.3** |

very low AUC scores, with the highest being a mere 0.28. These poor AUC values indicate that the discrimination ability of these similarity-based approaches is inadequate for effectively identifying relevant task vectors from the library. The precision-recall curves for similarity-based methods on other models are presented in Appendix G, further highlighting their suboptimal performance. In stark contrast, the trained retriever in our proposed design can achieve a remarkably high AUC of 0.96 (Figure 4a). This substantial improvement in retrieval performance underscores the benefits of our design, which effectively leverages learning-based techniques to intelligently retrieve and integrate relevant capabilities from the library.

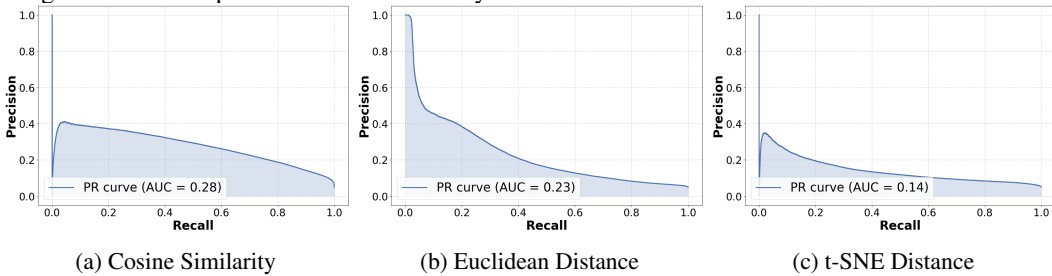

(a) Cosine Similarity  (b) Euclidean Distance  (c) t-SNE Distance

Figure 6: **Precision-Recall Curves for Similarity-based methods for Llama3**.

## 5.2 SELECTION AFTER THRESHOLD FILTERING

In Table 5, we investigate the impact of the number of selected states $n$ when the top1 similarity score reaches the threshold, as well as the effect of using Dynamic Top-K Thresholding (DTT) or not as Section 3.3 mentioned. The results show that selecting $n = 10$ and using Dynamic Top-K Thresholding achieves the best performance on the validation set. Choosing fewer vector quantities (e.g., n=5) would limit the method's potential, while selecting too many (e.g., n=15) could introduce irrelevant noise, thereby degrading performance. Dynamic Top-K adaptively sets the similarity threshold to ensure that only sufficiently relevant vectors are utilized. This ablation study highlights the rationale and effectiveness of our design choices.

Table 5: **Ablation study** on DDT on valid for Llama3-8B.

| | TP |
|---|---|
| **zs-shot** | 33.62 |
| $n$**=5** | 44.71 |
| $n$**=15** | 45.02 |
| $n$**=10,w/o DTT** | 45.17 |
| $n$**=10** | **45.29** |

## 6 CONCLUSION

In this paper, we explore the vision of eliciting and harnessing the potential of large language models' inherent capabilities when adapting to new tasks, akin to in-context learning (ICL), while maintaining efficiency and flexibility. We propose ELICIT, a novel framework consisting of two key modules: Build Capability Library and Dynamic Capability Elicitation. ELICIT achieves consistent improvements across diverse tasks, input formats, and model architectures. Our results show that ELICIT not only has the potential to harness models' latent abilities without introducing substantial additional computational cost, but also advances language models' performance, versatility, adaptability, and scalability.

ACKNOWLEDGEMENT

This work was supported in part by the National Science and Technology Major Project (No. 2022ZD0115101), Research Center for Industries of the Future (RCIF) at Westlake University, Westlake Education Foundation, and Westlake University Center for High-performance Computing.

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

## A INTERVENTION STRATEGIES

As mentioned in Section 3.2, we choose $\tilde{\mathbf{h}}_l = \mathbf{h}_l + \alpha \cdot \boldsymbol{\theta}_l$ as the intervention strategy, where $\alpha$ is a scaling factor that controls the intervention strength. We observe the performance and cross-entropy loss across a diverse set of 20 tasks by varying $\alpha$. The results for Mistral, Mamba, and Pythia are shown in Figures 7, 8, and 9, respectively. The results reveal a similar trade-off between task performance and language modeling capability as the intervention strength increases. Among the strategies tested, the additive approach consistently demonstrates superior performance across a wide range of tasks while minimizing degradation in language modeling ability. Across different models, $\alpha$ can be set to 2.0 to achieve a good balance between task performance and language modeling capability.

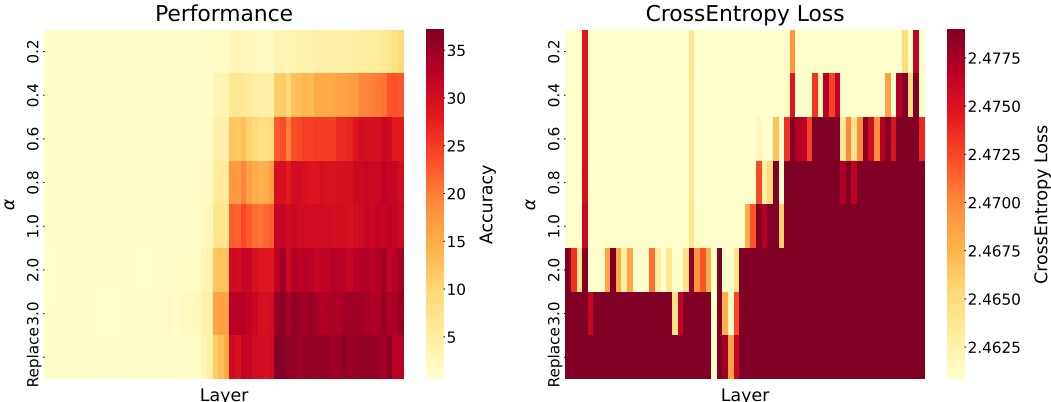

Figure 7: Varying intervention strengths affect accuracy and cross-entropy loss in Mamba on valid set of 20 tasks across different layer.

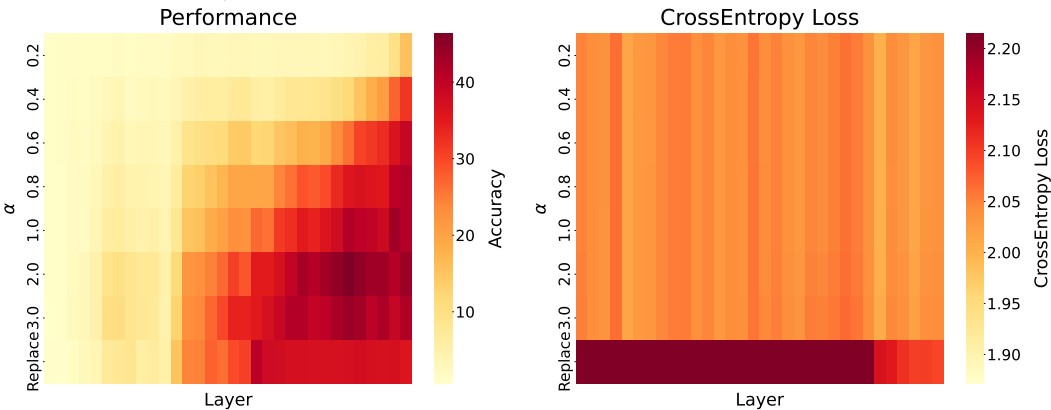

Figure 8: Varying intervention strengths affect accuracy and cross-entropy loss in Mistral on valid set of 20 tasks across different layer.

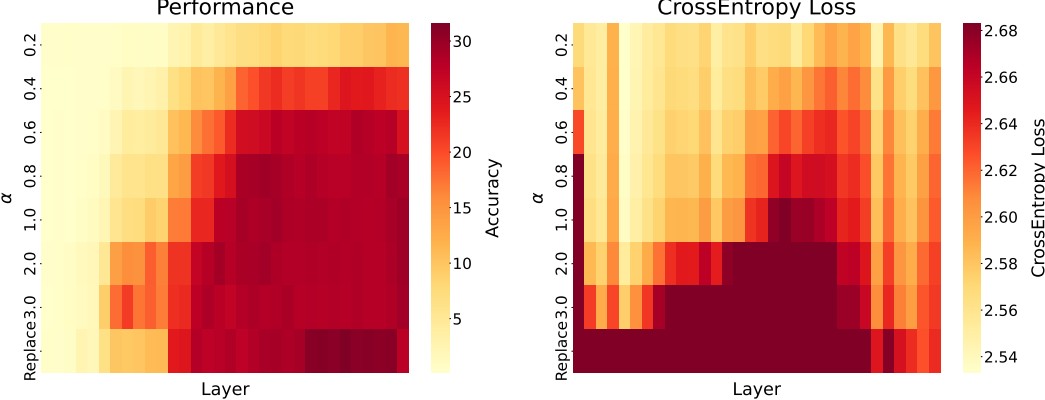

Figure 9: Varying intervention strengths affect accuracy and cross-entropy loss in Pythia on valid set of 20 tasks across different layer.

## A.1 POSSIBLE INTERPRETATION OF TRADE-OFF

The trade-off between intervention strength and language modeling performance can be explained through neural circuit interactions. Prior work suggests ICL and general language modeling operate with different circuits (Chan et al., 2022; Olsson et al., 2022; Singh et al., 2024). We hypothesize that stronger interventions redirect activation patterns from pretrained language modeling circuitry towards ICL-specific patterns, creating tension between these distinct computational paths. As inter-

vention strength increases, activations appear to deviate further from their pretrained configurations, potentially explaining the degraded performance on general language modeling tasks like Wiki-Text. This observation suggests that optimal intervention strength requires careful balancing rather than maximization. Further investigation of these circuit-level understanding remains an important direction for future research.

# B  BEST LAYER FOR DIFFERENT TASKS.

As mentioned in Section 3.2, we identify the optimal intervention layer for different tasks. Figures 10, 11, and 12 illustrate that the optimal intervention layer varies significantly across tasks. For instance, the ARC challenge task achieves the best performance when intervening in the middle layers, while the GLUE SST2 task performs best when intervening in the later layers for the Llama-3 model. Furthermore, intervening at different layers leads to substantial performance variations. Therefore, instead of fixing the intervention layer as in Todd et al. (2023), we propose a dynamic layer selection approach to identify the optimal intervention layer for each task.

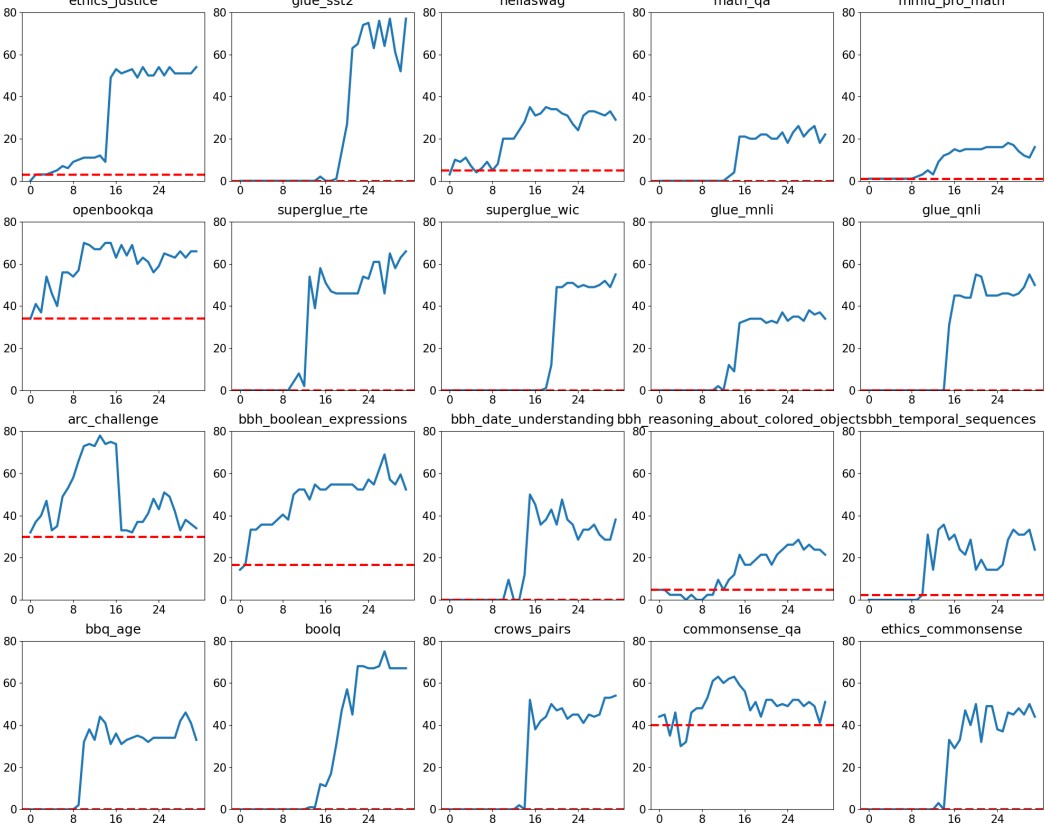

Figure 10: Performance distribution varying intervention layer on Llama3 8B in 20 tasks on Valid Set.

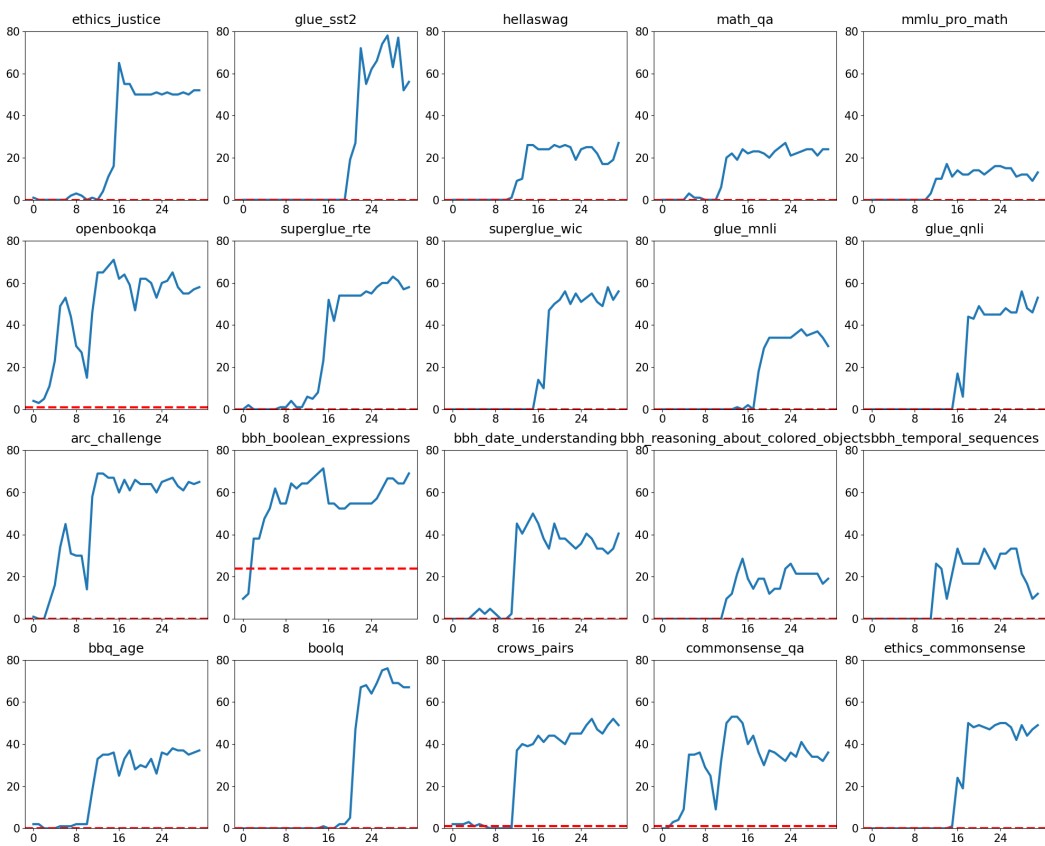

Figure 11: Performance distribution varying intervention layer on Mistral in 20 tasks on Valid Set.

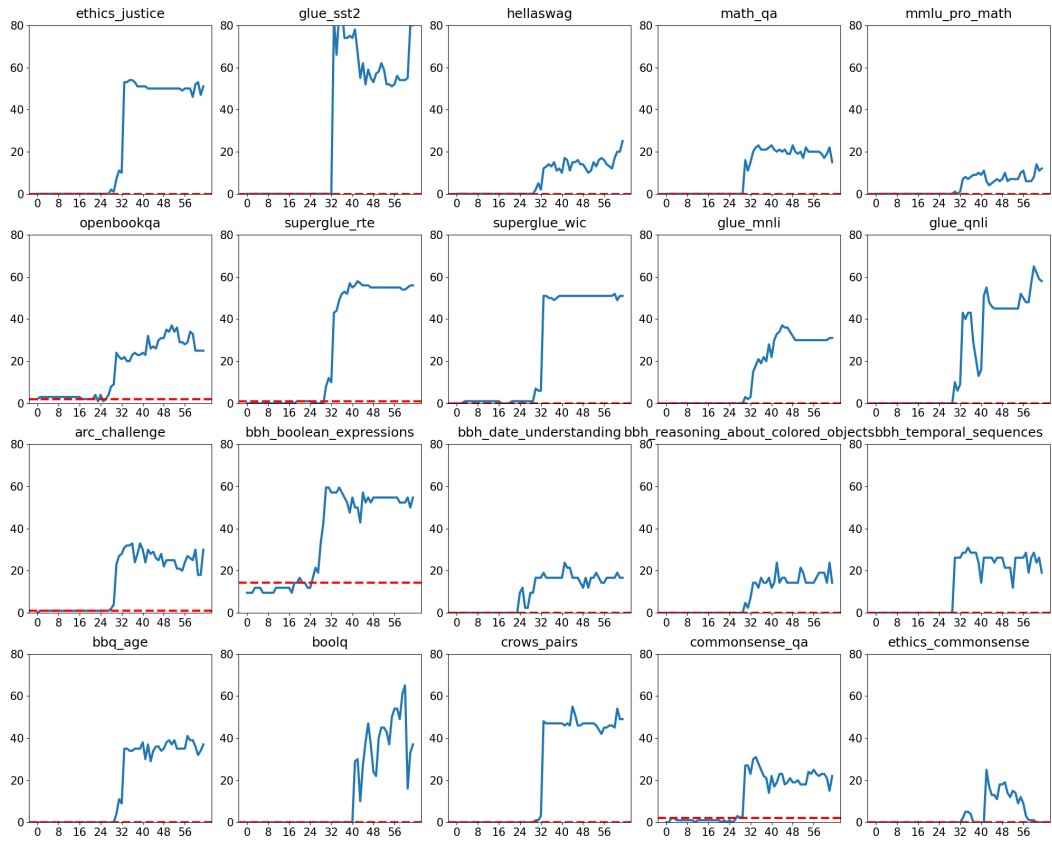

Figure 12: Performance distribution varying intervention layer on Mamba in 20 tasks on Valid Set.

## C    DATA CONSTRUCTION FOR RETRIEVER

As mentioned in Section 3.3, we train a retriever on the constructed data. To train the retriever as a classifier, we construct data pairs to determine whether a task-specific prompt and an ICL prompt belong to the same task. For each task and each template, we sample two examples from the validation set. We create positive and negative pairs, where each pair consists of a task-specific prompt and an ICL prompt from the library. Negative examples are formed by randomly pairing task-specific prompts with ICL prompts from different tasks. We balance the data distribution across different tasks, ensuring that each task has an equal number of positive and negative pairs.

## D    RECALL SWEEP

As mentioned in Section 3.3, we sweep the threshold determined by recall, and as shown in Figure 13, the results reveal that a recall of 0.8 provides an optimal balance between accuracy and recall for our pipeline across Pythia, Mamba, and Mistral models.

We observe a similar increasing trend in performance as recall increases for all three models. At lower recall values, the intervene accuracy is close to the zero-shot accuracy, indicating that the retrieved prompts may not be relevant to the task. As recall increases, the intervene accuracy improves significantly, demonstrating the effectiveness of the proposed approach in selecting appropriate prompts for intervention.

Based on these observations, we choose a recall value of 0.8 to determine the threshold for filtering prompts across different models, as it strikes a balance between maximizing accuracy and maintaining a reasonable recall level.

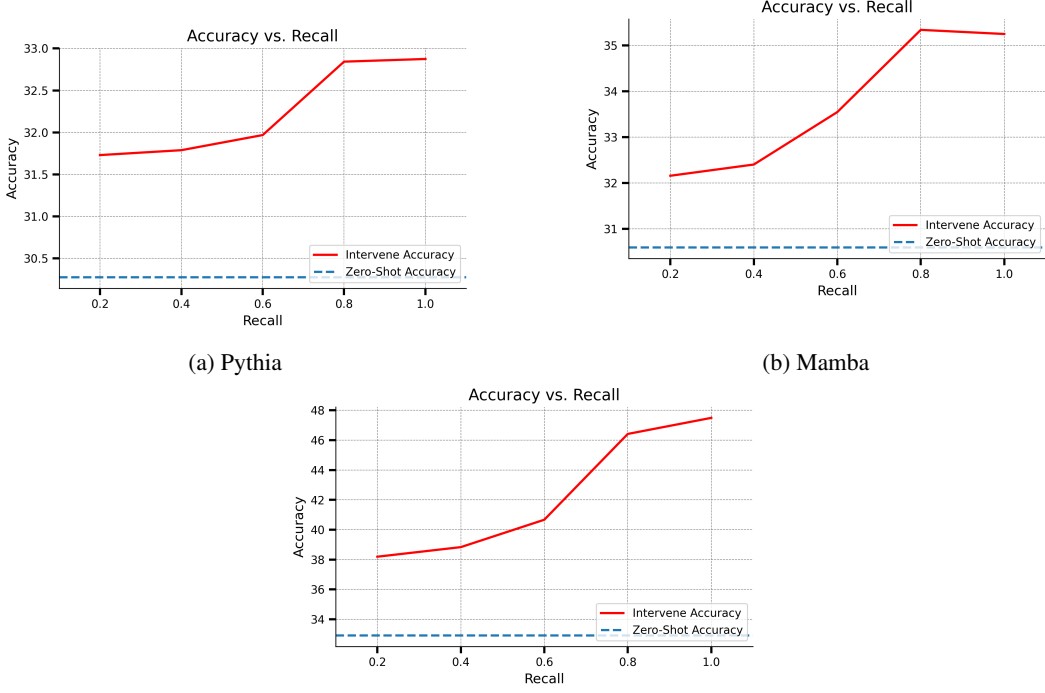

(a) Pythia                 (b) Mamba

(c) Mistral

Figure 13: Accuracy vs. Recall curves for Pythia, Mamba, and Mistral models, illustrating the performance trade-off at different recall levels.

# E    EVALUATION SETTING

## E.1    ICL SETTING

As 4.1 states, we first primarily implement ELICIT on the traditional In-Context Learning (ICL) setting. We find that zero-shot Large Language Models (LLMs) cannot answer properly with contextual guidance. Although ELICIT works on such a traditional ICL setting, as shown in Table 6 and 7, the zero-shot accuracy is almost 0, which is not plausible to evaluate model's performance. Therefore, we think it's not fair to augment model performance on such traditional zero-shot queries.

Table 6: Performance of ELICIT across model and tasks in ICL setting.

| model | | #Tokens | NLU | Reasoning | Knowledge | Math | Safety | Avg. |
|---|---|---|---|---|---|---|---|---|
| **Llama** | 16-shot | 1553.4 ± 2.8 | 60.3 ± 1.2 | 55.6 ± 0.2 | 69.2 ± 2.0 | 27.0 ± 0.0 | 60.9 ± 0.5 | 54.6 ± 0.6 |
| | BM25 | 1799.2 ± 26.8 | 59.4 ± 0.3 | 54.7 ± 0.2 | 66.7 ± 0.2 | 30.3 ± 0.9 | 55.4 ± 1.2 | 53.3 ± 0.5 |
| | **Zero-shot** | 87.6 ± 0.8 | 0.0 ± 0.0 | 21.6 ± 0.2 | 26.2 ± 0.2 | 0.0 ± 0.0 | 3.1 ± 0.6 | 10.2 ± 0.1 |
| | **Ours** | 87.6 ± 0.8 | **45.5 ± 1.5** | **45.2 ± 0.2** | **57.6 ± 0.1** | **12.7 ± 0.9** | **43.6 ± 2.0** | **40.9 ± 0.1** |
| **Mistral** | 16-shot | 1779.0 ± 3.4 | 59.5 ± 1.5 | 51.7 ± 0.7 | 69.4 ± 1.6 | 24.0 ± 3.7 | 62.2 ± 1.8 | 53.4 ± 1.3 |
| | BM25 | 2045.0 ± 27.9 | 57.5 ± 1.3 | 51.8 ± 0.5 | 66.2 ± 2.5 | 23.8 ± 1.6 | 59.2 ± 1.3 | 51.7 ± 0.6 |
| | **Zero-shot** | 100.9 ± 1.7 | 0.1 ± 0.1 | 22.6 ± 0.8 | 16.9 ± 0.9 | 0.7 ± 0.5 | 2.7 ± 0.4 | 8.6 ± 0.1 |
| | **Ours** | 100.9 ± 1.7 | **31.8 ± 0.5** | **44.4 ± 0.6** | **48.0 ± 0.7** | **17.0 ± 3.1** | **41.8 ± 1.0** | **36.6 ± 0.1** |
| **Pythia** | 16-shot | 1581.1 ± 0.3 | 52.7 ± 1.7 | 21.7 ± 1.0 | 13.7 ± 1.2 | 12.2 ± 1.5 | 34.1 ± 0.2 | 26.8 ± 0.2 |
| | BM25 | 1848.6 ± 26.4 | 47.9 ± 1.7 | 20.8 ± 0.8 | 20.2 ± 0.8 | 12.3 ± 2.8 | 36.4 ± 1.0 | 27.5 ± 0.3 |
| | **Zero-shot** | 88.3 ± 1.5 | 0.2 ± 0.0 | 7.7 ± 0.3 | 3.2 ± 0.4 | 0.3 ± 0.1 | 2.0 ± 0.0 | 2.7 ± 0.1 |
| | **Ours** | 88.3 ± 1.5 | **46.3 ± 0.8** | **23.2 ± 1.1** | **11.1 ± 0.8** | **15.2 ± 2.0** | **36.9 ± 3.0** | **26.5 ± 0.8** |
| **Mamba** | 16-shot | 1581.1 ± 1.3 | 40.4 ± 1.1 | 31.9 ± 1.1 | 34.2 ± 1.0 | 15.0 ± 2.9 | 40.9 ± 2.1 | 32.5 ± 0.3 |
| | BM25 | 1848.6 ± 26.4 | 43.7 ± 1.9 | 31.4 ± 0.0 | 25.2 ± 1.9 | 14.7 ± 2.1 | 38.3 ± 0.8 | 30.7 ± 0.5 |
| | **Zero-shot** | 88.3 ± 1.5 | 0.2 ± 0.2 | 12.3 ± 0.2 | 0.6 ± 0.2 | 0.0 ± 0.0 | 5.1 ± 1.5 | 3.6 ± 0.3 |
| | **Ours** | 88.3 ± 1.5 | **33.7 ± 0.6** | **26.5 ± 0.6** | **25.6 ± 1.1** | **16.3 ± 2.2** | **39.2 ± 0.5** | **28.3 ± 0.2** |

Table 7: Unseen task for ICL Setting

| | | # Tokens | GLUE COLA | BBQ Religion | Deepmind | MMLU-Psychology | BBH-five-objects | Avg |
|---|---|---|---|---|---|---|---|---|
| **Llama** | BM25 | 1684.3 ± 3.3 | 33.3 ± 1.9 | 69.3 ± 2.4 | 32.0 ± 1.4 | 83.7 ± 0.5 | 35.0 ± 0.0 | 50.7 ± 0.1 |
| | **Zero-shot** | 76.5 ± 0.2 | 0.7 ± 0.5 | 5.3 ± 2.4 | 0.0 ± 0.0 | 62.7 ± 0.5 | 0.0 ± 0.0 | 13.7 ± 0.7 |
| | **Ours** | 76.5 ± 0.2 | **1.3 ± 0.5** | **24.3 ± 3.8** | **19.3 ± 2.4** | **66.0 ± 0.0** | **5.0 ± 0.0** | **23.2 ± 1.1** |
| **Mistral** | BM25 | 1913.3 ± 20.0 | 27.0 ± 1.6 | 69.0 ± 1.4 | 28.3 ± 3.9 | 79.7 ± 1.2 | 26.2 ± 0.0 | 46.1 ± 0.6 |
| | **Zero-shot** | 85.5 ± 0.8 | **1.7 ± 1.2** | 1.0 ± 0.8 | 1.3 ± 1.2 | 35.3 ± 1.2 | 0.0 ± 0.0 | 7.9 ± 0.3 |
| | **Ours** | 85.5 ± 0.8 | 1.3 ± 1.2 | **18.3 ± 2.1** | **20.0 ± 0.8** | **54.7 ± 1.2** | **10.0 ± 0.0** | **20.9 ± 1.1** |
| **pythia** | BM25 | 1747.6 ± 20.2 | 13.7 ± 1.9 | 33.7 ± 1.7 | 20.3 ± 2.6 | 18.7 ± 1.7 | 6.2 ± 0.0 | 18.5 ± 1.2 |
| | **Zero-shot** | 78.1 ± 0.7 | **43.0 ± 2.9** | 0.0 ± 0.0 | 0.3 ± 0.5 | 3.3 ± 0.5 | 0.0 ± 0.0 | 9.3 ± 0.7 |
| | **Ours** | 78.1 ± 0.7 | 37.0 ± 2.4 | **14.7 ± 4.6** | **15.0 ± 0.8** | **13.0 ± 1.4** | **6.2 ± 0.0** | **17.2 ± 0.8** |
| **Mamba** | BM25 | 1747.6 ± 20.2 | 36.7 ± 1.2 | 33.3 ± 2.6 | 25.3 ± 4.5 | 25.3 ± 0.5 | 21.2 ± 0.0 | 28.4 ± 1.1 |
| | **Zero-shot** | 78.1 ± 0.7 | **19.7 ± 2.5** | 0.0 ± 0.0 | 0.0 ± 0.0 | 0.0 ± 0.0 | 0.0 ± 0.0 | 3.9 ± 0.5 |
| | **Ours** | 78.1 ± 0.7 | 18.0 ± 1.4 | **19.3 ± 3.4** | **13.7 ± 3.8** | **13.0 ± 0.8** | **7.5 ± 0.0** | **14.3 ± 1.1** |

## E.2 TASK SPECIFIC PROMPT

To ensure fair comparisons in zero-shot scenarios, we prepend task-specific prompts before each test query. These task-specific prompts are manually crafted following guidelines from `lm-harness` (Gao et al., 2023b) and the `chain-of-thought-hub` [3]. The complete set of prompts used is provided in Figure 8.

Table 8: Task Specific Prompts for Various Tasks

| Task | Prompts |
|---|---|
| bbh_date_understanding | • Infer the date from context. Finish your answer with 'X' where X is the correct letter choice.

Question: {input}
• Determine the date based on contextual clues. End your response with 'X', where X represents the correct option.

Question: {input}
• Use the given context to deduce the date. Conclude your answer with 'X', X being the right letter choice.

Question: {input} |
| bbh_boolean_expressions | • Evaluate the result of a random Boolean expression.

Question: {input}
• Calculate the outcome of a given Boolean expression.

Question: {input}
• Determine the result of the provided Boolean logic statement.

Question: {input} |

*Continued on next page*

---

[3]https://github.com/FranxYao/chain-of-thought-hub.git

**Prompts for Various Tasks (Continued)**

| Task | Prompts |
|---|---|
| bbh_date_understanding | <ul><li>Infer the date from context. Finish your answer with 'X' where X is the correct letter choice.

Question: {input}</li><li>Determine the date based on contextual clues. End your response with 'X', where X represents the correct option.

Question: {input}</li><li>Use the given context to deduce the date. Conclude your answer with 'X', X being the right letter choice.

Question: {input}</li></ul> |
| bbh_boolean_expressions | <ul><li>Evaluate the result of a random Boolean expression.

Question: {input}</li><li>Calculate the outcome of a given Boolean expression.

Question: {input}</li><li>Determine the result of the provided Boolean logic statement.

Question: {input}</li></ul> |
| mmlu_pro_math | <ul><li>The following are multiple choice questions (with answers) about math. Finish your answer with 'X' where X is the correct letter choice.

Question: {input}</li><li>Below are multiple-choice math questions. Conclude your response with 'X', X being the correct option.

Question: {input}</li><li>Answer these math multiple-choice questions. Answer with 'X', where X is the right letter choice.

Question: {input}</li></ul> |

*Continued on next page*

**Prompts for Various Tasks (Continued)**

| Task | Prompts |
|---|---|
| superglue_rte | • {input}
Question: Is the hypothesis correct based on the premise? Answer True or False.
• {input}
Question: Based on the premise, is the hypothesis accurate? Respond with True or False.
• {input}
Question: Does the premise support the hypothesis? Answer True or False. |
| bbh_temporal_sequences | • Answer questions about which times certain events could have occurred. Finish your answer with 'X' where X is the correct letter choice.

Q: {input}
• Determine possible occurrence times for specific events. Answer with 'X', X being the correct option.

Q: {input}
• Identify when certain events could have happened. Conclude with 'X', where X is the right letter choice.

Q: {input} |
| bbh_reasoning_about_colored_objects | • Answer extremely simple questions about the colors of objects on a surface. Finish your answer with 'X' where X is the correct letter choice.

Q: {input}
• Respond to basic questions about object colors on a surface. Answer with 'X', X being the correct option.

Q: {input}
• Address simple queries regarding the colors of items on a surface. Conclude with 'X', where X is the right letter choice.

Q: {input} |

*Continued on next page*

**Prompts for Various Tasks (Continued)**

| Task | Prompts |
|---|---|
| ethics_commonsense | <ul><li>{input}
Question: Is this wrong? Answer Yes or No.</li><li>{input}
Question: Is this ethically incorrect? Respond with Yes or No.</li><li>{input}
Question: Does this violate ethical norms? Answer Yes or No.</li></ul> |
| ethics_justice | <ul><li>{input}
Question: Would most people believe this reasonable or unreasonable to say? Answer Yes or No.</li><li>{input}
Question: Would the majority consider this statement reasonable or unreasonable? Answer Yes or No.</li><li>{input}
Question: Is this statement generally perceived as reasonable or unreasonable? Respond with Yes or No.</li></ul> |
| glue_mnli | <ul><li>{input} True, False or Neither?</li><li>{input} Answer selecting from: True, False, or Neither?</li><li>{input} Choose one as answer: True, False, or Neither?</li></ul> |
| glue_sst2 | <ul><li>{input}
Question: Is this sentence positive or negative? Answer positive or negative.</li><li>{input}
Question: Does this sentence convey a positive or negative sentiment? Answer positive or negative.</li><li>{input}
Question: Is the tone of this sentence positive or negative? Respond with positive or negative.</li></ul> |
| glue_qnli | <ul><li>{input}
Question: Does this response answer the question? Answer Yes or No.</li><li>{input}
Question: Is the response answer to the question? Answer Yes or No.</li><li>{input}
Question: Does the given response address the question? Respond with Yes or No.</li></ul> |

*Continued on next page*

**Prompts for Various Tasks (Continued)**

| Task | Prompts |
|------|---------|
| hellaswag | • The following are multiple choice questions (with answers) about commonsense NLI. Finish your answer with 'X' where X is the correct letter choice.

Question: {input}
• The following are multiple-choice questions about commonsense natural language inference. Answer with 'X', X being the correct option.

Question: {input}
• Answer these multiple-choice questions on commonsense language understanding. Conclude with 'X', where X is the right letter choice.

Question: {input} |
| crows_pairs | • {input}
Question: Which sentence is more stereotypical? Answer A or B.
• {input}
Question: Which sentence contains more stereotypical content? Answer A or B.
• {input}
Question: Identify the sentence with stronger stereotypical elements. Respond with A or B. |
| boolq | • {input}
Answer True or False.
• {input}
Respond with True or False.
• {input}
Is this statement correct? Answer True or False. |

**Prompts for Various Tasks (Continued)**

| Task | Prompts |
|---|---|
| math_qa | <ul><li>The following are multiple choice questions (with answers) about math word problem. Finish your answer with 'X' where X is the correct letter choice.

Question: {input}</li><li>Below are multiple-choice questions about math word problems. Answer with 'X', X being the correct option.

Question: {input}</li><li>Solve these multiple-choice math word problems. Conclude with 'X', where X is the right letter choice.

Question: {input}</li></ul> |
| superglue_wic | <ul><li>{input}
Question: Is the word used in the same way in the two sentences above? Answer Yes or No.</li><li>{input}
Question: Is the word used similarly in both sentences above? Respond with Yes or No.</li><li>{input}
Question: Does the word have the same meaning in the two given sentences? Answer Yes or No.</li></ul> |
| openbookqa | <ul><li>The following are multiple choice questions (with answers) about multi-step reasoning. Finish your answer with 'X' where X is the correct letter choice.

Question: {input}</li><li>The following are multiple-choice questions testing multi-step reasoning. Answer with 'X', X being the correct option.

Question: {input}</li><li>Answer these multiple-choice questions involving multi-step logical thinking. Conclude with 'X', where X is the right letter choice.

Question: {input}</li></ul> |

*Continued on next page*

**Prompts for Various Tasks (Continued)**

| Task | Prompts |
|---|---|
| commonsense_qa | • The following are multiple choice questions (with answers) about commonsense knowledge reasoning. Finish your answer with 'X' where X is the correct letter choice.

Question: {input}
• Below are multiple-choice questions about commonsense reasoning. Answer with 'X', X being the correct option.

Question: {input}
• Respond to these multiple-choice questions on commonsense knowledge. Conclude with 'X', where X is the right letter choice.

Question: {input} |
| bbq_age | • The following are multiple choice questions (with answers) about social bias on age. Finish your answer with 'X' where X is the correct letter choice.

Question: {input}
• The following are multiple-choice questions about age-related social bias. Answer with 'X', X being the correct option.

Question: {input}
• Answer these multiple-choice questions on social bias regarding age. Conclude with 'X', where X is the right letter choice.

Question: {input} |
| arc_challenge | • The following are multiple choice questions (with answers) about science. Finish your answer with 'X' where X is the correct letter choice.

Question: {input}
• Below are multiple-choice science questions. Answer with 'X', X being the correct option.

Question: {input}
• Respond to these multiple-choice questions on scientific topics. Conclude with 'X', where X is the right letter choice.

Question: {input} |

**Prompts for Various Tasks (Continued)**

| Task | Prompts |
|---|---|
| glue_cola | • {input}
Question: Does this sentence make sense? Answer Yes or No.
• {input}
Question: Is this sentence logically coherent? Respond with Yes or No.
• {input}
Question: Evaluate if this sentence is meaningful. Reply with Yes or No. |
| bbh_logical_deduction_five_objects | • A logical deduction task which requires deducing the order of a sequence of objects. Finish your answer with 'X' where X is the correct letter choice.

Question: {input}
• This challenge involves logically determining the sequence of a set of objects. Conclude your response with 'X', where X is the appropriate letter option.

Question: {input}
• In this logical reasoning exercise, deduce the correct order of a series of objects. End your answer with 'X', X being the right letter choice.

Question: {input} |
| mmlu_high_school_psychology | • The following are multiple choice questions (with answers) about high school psychology. Finish your answer with 'X' where X is the correct letter choice.

Question: {input}
• Below are multiple-choice questions testing high school level psychology knowledge. Conclude your response with 'X', X representing the correct option.

Question: {input}
• These questions assess understanding of high school psychology concepts. End your answer with 'X', where X is the letter of the correct choice.

Question: {input} |

*Continued on next page*

**Prompts for Various Tasks (Continued)**

| Task | Prompts |
|---|---|
| bbq_religion | • The following are multiple choice questions (with answers) about social bias on religion. Finish your answer with 'X' where X is the correct letter choice.

Question: {input}
• Here are multiple-choice questions addressing social biases related to religion. Conclude your answer with 'X', X being the correct letter option.

Question: {input}
• These questions explore social biases in the context of religion. End your response with 'X', where X represents the right letter choice.

Question: {input} |
| deepmind | • The following are multiple choice questions (with answers) about algebraic word problems. Finish your answer with 'X' where X is the correct letter choice.

Question: {input}
• Below are multiple-choice questions testing algebraic word problem solving skills. Conclude your answer with 'X', X being the correct option letter.

Question: {input}
• These questions assess your ability to solve algebraic word problems. End your response with 'X', where X is the letter of the right choice.

Question: {input} |

# F ADAPTIVE ELICITATION

As mentioned in Section 4.3, we show that when provided the library with only math-related task vectors, performance shows a significant improvement on the math domain while retaining or slightly improving in other domains for Mistral. Figure 14 illustrates similar results on other models such as Mamba, Pythia, and Llama3.

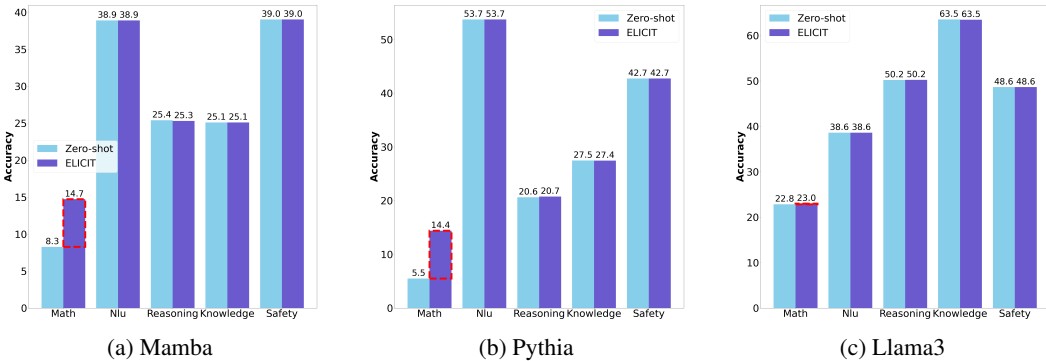

Figure 14: Performance on ELICITacross different domains when the li-brary only contains math-related taskvectors on Mamba, Pythia, and Llama3.

## G    SIMILARITY-BASED RETRIEVE METHOD

Section 5.1 demonstrates the poor precision-recall performance of similarity-based retrieval methods on the Llama3 model. Figure 15 presents the Precision-Recall curves for Mistral and Mamba under different similarity-based approaches, which also exhibit poor results. In contrast, our proposed retrieval module achieves significantly higher precision and recall across all models. This highlights the effectiveness of our method in accurately retrieving relevant task vectors to support different tasks.

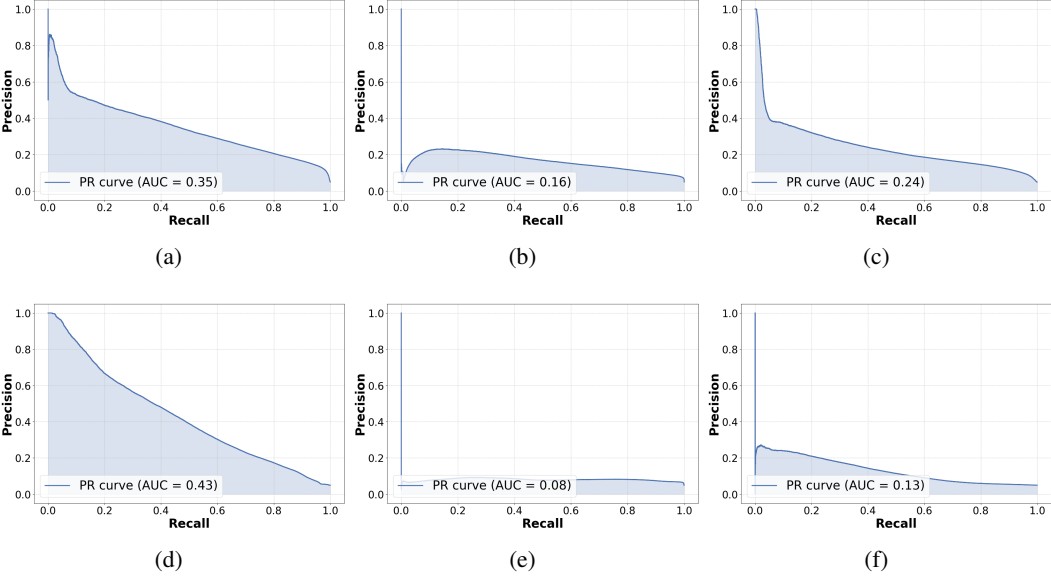

Figure 15: Precision-Recall Curves for Similarity-based Prompt Retrieval Methods on Mistral and Mamba Models. Top Row: Mistral, Bottom Row: Mamba. From Left to Right: Cosine Similarity, t-SNE, and Distance-based Methods.

## H    ORTHOGONAL AUGMENTATION IN UNSEEN TASKS.

As shown in Section 4.5, ELICIT demonstrates its plug-and-play capability by seamlessly integrating with existing methods such as BM25 Retrieval for in-domain tasks. Table 9 showcases ELICIT combined with BM25 in unseen tasks, combined with BM25 for unseen tasks, where we can observe performance improvements across various models and tasks. This highlights the versatility and effectiveness of ELICIT in augmenting existing methods and tasks.

Table 9: ELICIT combined with BM25 on unseen tasks. Improvements are concentrated in smaller models (Pythia, Mamba), while larger models exhibit task-specific trade-offs (e.g., Llama3 shows gains in Deepmind but declines in GLUE-COLA).

| Model | | GLUE COLA | BBQ Religion | Deepmind | MMLU-Psychology | BBH-five-objects | Avg |
|---|---|---|---|---|---|---|---|
| Llama | BM25 | **55.4 ± 1.0** | **64.6 ± 1.3** | **30.7 ± 1.7** | **83.0 ± 0.1** | **48.3 ± 0.0** | **56.4 ± 0.4** |
| | BM25+ELICIT | 47.6 ± 2.2 | 60.6 ± 1.0 | 26.4 ± 1.0 | 81.4 ± 0.8 | 44.4 ± 0.0 | 52.1 ± 0.3 |
| Mistral | BM25 | **44.4 ± 2.2** | **70.7 ± 0.7** | **26.6 ± 3.9** | **78.7 ± 1.1** | 25.7 ± 0.0 | **49.2 ± 0.3** |
| | BM25+ELICIT | 36.4 ± 1.1 | 59.4 ± 1.8 | 25.2 ± 1.6 | 70.5 ± 0.3 | **26.9 ± 0.0** | 43.7 ± 0.5 |
| Pythia | BM25 | 5.8 ± 1.0 | 19.1 ± 1.2 | **14.1 ± 1.2** | 4.7 ± 0.3 | 1.0 ± 0.0 | 8.9 ± 0.3 |
| | BM25+ELICIT | **7.3 ± 0.8** | **30.9 ± 3.3** | 14.0 ± 0.6 | **11.9 ± 0.6** | **3.5 ± 0.0** | **13.5 ± 0.7** |
| Mamba | BM25 | **48.1 ± 3.1** | 30.6 ± 1.1 | 21.6 ± 3.3 | 19.1 ± 0.9 | **25.8 ± 0.0** | **29.0 ± 0.9** |
| | BM25+ELICIT | 46.6 ± 1.7 | **30.9 ± 1.8** | **22.7 ± 0.6** | **22.7 ± 0.4** | 21.8 ± 0.0 | 28.9 ± 0.5 |

## I    DATASET SPLITS

We provide detailed information about our dataset curation and splitting strategies to ensure reproducibility. Our primary objective was to maintain robust evaluation capabilities while ensuring sufficient training data for ICL prompt construction. For datasets with pre-existing splits (ARC-Challenge, Ethics, GLUE, MathQA, OpenbookQA), we preserved the original partitioning. When handling datasets with only train-valid splits, we employed two approaches: for those with validation sets exceeding 350 samples (e.g., BoolQ, Hellaswag), we split the validation set into new validation and test sets at a 7:3 ratio; for those with smaller validation sets (e.g., CommonsenseQA), we divided the training set into new train and test sets (7:3). For test-only datasets, we implemented different strategies based on size: smaller datasets like BBH (250 samples) were split to ensure 128 samples for training and 80-100 samples for testing, with remaining samples allocated to validation. Larger test-only datasets (>1000 samples) such as MMLU-Pro-Math, BBQ, and Crows Pairs were split into train-valid-test sets at a 7:2:1 ratio. The same 7:2:1 split was applied to train-only datasets like SuperGLUE and DeepMind. This systematic approach ensures a minimum of 80 test samples for reliable evaluation metrics and at least 128 training samples for ICL prompt construction across all tasks.

## J    ANALYSIS OF ELICIT'S SELECTIVE ACTIVATION

We investigate why ELICIT can selectively activate capability in Figure 5 and the importance of this mechanism. Using a library containing only math-related task vectors on Mistral, we analyzed the number of chosen states per domain, shown in Table 10. Math-related tasks showed consistent high utilization (9.8 ± 0.1 chosen states), while other domains maintained minimal selection (approximately 0.0). This pattern confirms that ELICIT's performance improvements stem from its dynamic retrieval and selective activation of relevant capabilities.

Table 10: The average number of chosen numbers per domain per sample. The statistics come from Mistral when the capability library only contains math-related task vectors.

| | in-domain | | | | |
|---|---|---|---|---|---|
| | NLU | Reasoning | Knowledge | Math | Safety |
| chosen nums | 0.0 ± 0.0 | 0.1 ± 0.0 | 0.0 ± 0.0 | **9.8 ± 0.1** | 0.0 ± 0.0 |
| | Out-of-domain | | | | |
| | GLUE COLA | BBQ Religion | Deepmind | MMLU-Psychology | BBH-five-objects |
| chosen nums | 0.0 ± 0.0 | 0.0 ± 0.0 | **9.9 ± 0.1** | 0.0 ± 0.0 | 0.0 ± 0.0 |

We observed minor improvements in reasoning tasks, exemplified by this ARC Challenge case in Table 11. It demonstrates our pipeline's ability to selectively activate relevant capabilities based solely on query and handle unseen tasks flexibly, without requiring explicit task information.

Experiments forcing the application of top task vectors to all queries (Table 12), showed significant performance degradation in NLU and knowledge tasks, highlighting the importance of selective activation.

Table 11: A successful case from Arc-Challenge when capability library only contains math-related task vectors on Mistral.

| | |
|---|---|
| **input** | Below are multiple-choice science questions.Answer with 'X', X being the correct option.\n\nQuestion: An unbalanced equation for the reaction of methane gas (CH_{4}) with oxygen is shown below. CH_{4} + \\Box O_{2} ->2CO_{2} + 4H_{2}O How many molecules of oxygen gas (O_{2}) are needed to properly balance this equation?\nOptions:\nA. 1\nB. 2\nC. 3 \nD. 4\nAnswer: |
| **chosen task vectors** | 10 task vectors from MathQA |
| **Original Output** | B |
| **ELICIT Output** | **D (correct)** |

Table 12: The results of forcibly applying the top task vectors for each query. The experiments were conducted on Mistral. Domains with degraded performance are marked in bold.

| | nlu | reasoning | knowledge | math | safety |
|---|---|---|---|---|---|
| **Zero-shot** | 28.8 | 27.4 | 58.8 | 4.0 | 42.2 |
| **ELICIT** | **15.7** | 31.4 | **47.8** | 18.3 | 53.1 |

These experimental results demonstrate that ELICIT's performance improvement stems from its selective activation mechanism and the importance of selectively using only task-relevant vectors to dynamically activate capabilities.

We analyzed the usage frequency of task vectors in the capability library, which contains 20 distinct task vector types. The analysis was performed on Pythia-6.9B while evaluating 25 tasks in total: 20 in-domain tasks and 5 out-of-domain tasks. Our findings confirmed that all 20 task vector types in the library were utilized during the evaluation.

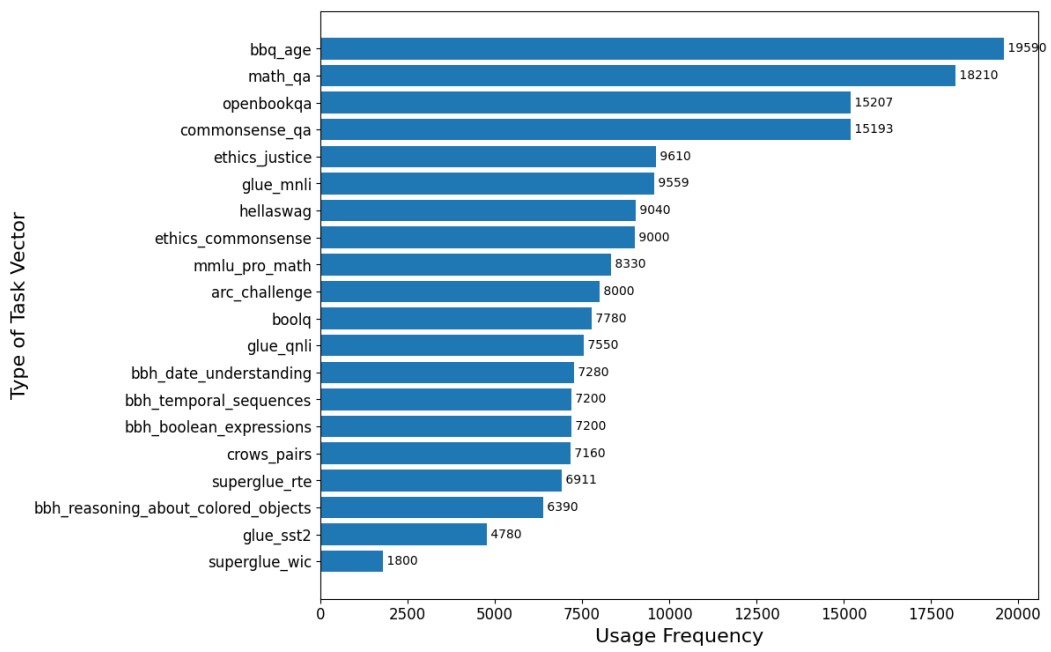

Figure 16: Usage Frequency Distribution of Different Types of Task Vectors Across all In-Domain and OOD (Out-of-Domain) Samples. The results is based on Pythia-6.9B.

## K MORE MODELS AND TASKS ON ELICIT

Beyond our primary experiments, we evaluate the scalability and generalizability of ELICIT across larger language models and more challenging tasks. As shown in Table 13, ELICIT maintains its performance advantages when applied to more base models. Furthermore, Table 14 demonstrates that ELICIT achieves consistent improvements across a diverse set of complex tasks, validating its effectiveness and versatility.

We further explore the applicability of ELICIT to instruction-tuned models, with preliminary results shown in Table 15. While this initial experiment suggest the potential compatibility of ELICIT with instruction-tuned models, several challenges remain. Instruction-tuned models exhibit heightened sensitivity to prompts and instructions (Sun et al., 2023; Gao et al., 2023a), necessitating more investigation and analysis. Key challenges include identifying effective task vectors for in-context learning (ICL) and developing robust methods for zero-shot performance evaluation. We leave the comprehensive adaptation of ELICIT for instruction-tuned models as promising future work.

Table 13: Performance of ELICIT on more different models. ELICIT are effective for larger models.

| | | Length | nlu | reasoning | knowledge | math | safety | avg |
|---|---|---|---|---|---|---|---|---|
| **Pythia-6.9B** | zs | 109.8 ± 1.5 | 37.6 ± 0.4 | 16.1 ± 0.5 | 17.4 ± 0.6 | 5.9 ± 0.7 | 31.7 ± 0.5 | 21.8 ± 0.1 |
| | ELICIT | 109.8 ± 1.5 | **38.7 ± 1.4** | **28.1 ± 0.5** | **27.9 ± 1.0** | **18.2 ± 2.6** | **47.8 ± 2.0** | **32.2 ± 0.7** |
| **Pythia-12B** | zs | 109.8 ± 1.5 | 34.7 ± 0.6 | 20.7 ± 0.2 | 18.1 ± 0.6 | 7.9 ± 1.7 | 34.6 ± 0.6 | 23.2 ± 0.2 |
| | ELICIT | 109.8 ± 1.5 | **38.5 ± 0.5** | **29.7 ± 0.7** | **29.8 ± 0.6** | **17.5 ± 2.1** | **46.8 ± 0.2** | **32.5 ± 0.5** |
| **Llama3-70B** | zs | 101.1 | 50.9 | 66.8 | 59.7 | 37.6 | 44.2 | 51.8 |
| | ELICIT | 101.1 | **55.9** | **80.5** | **84.6** | **52.4** | **67.4** | **68.2** |

Table 14: The results of ELICIT on GSM8K and MMLU-Professional-Law on Llama3-8B. GSM8K is as in-domain task and MMLU-Profeesional-Law is out-of-domain.

| | GSM8K | MMLU-Professional-Law |
|---|---|---|
| zs | 30.44 | 31.67 |
| **ELICIT** | **32.44** | **41.11** |

Table 15: The preliminary experiment of ELICIT on Llama3-8B-Instruct.

| | nlu | reasoning | knowledge | safety | avg |
|---|---|---|---|---|---|
| zs | 45.0 | 4.9 | 31.9 | 42.5 | 31.1 |
| **ELICIT** | **52.7** | **36.2** | **70.9** | **49.0** | **52.2** |

## L DIVERSITY-OPTIMIZAED CAPABILITY LIBRARY

We conduct an experiment on maximizing the diversity of prompts in the given capability library. Instead of random demonstration selection, we construct a new capability library of diversity-optimized prompts as described in Su et al. (2022).

Spefically, we used Sentence-BERT to generate embeddings by averaging the resulting vectors over the words in each text input. For each task, after computing embeddings for all training data, we implemented an iterative approach to find diverse examples to construct ICL prompts. Starting with a random example, we selected examples that maximized the distance from previously chosen examples in each iteration. We then conducted a new capability library using these more diverse ICL prompts.

As shown in Table 16, the diversity-optimized prompts yielded mixed results. Compared to the original ELICIT, while performance improved in reasoning (+1.1%), math (+0.5%) and NLU tasks (+4.5%), there was a decline in Knowledge (-5.9%) and Safety (-2.3%) ability.

This result suggests the potential for future work to improve our pipeline by enhancing the quality of task vectors through better demonstration selection methods.

Table 16: The comparison of ELICIT using different capability library based on different ICL prompts. The experiments are conducted on Llama3-8B.

|  | NLU | Reasoning | Knowledge | Math | Safety | Avg. |
|---|---|---|---|---|---|---|
| Zero-shot | 32.2 ± 1.2 | 32.9 ± 0.2 | 42.5 ± 1.2 | 14.0 ± 1.0 | 35.5 ± 1.2 | 31.4 ± 0.7 |
| ELICIT | 38.1 ± 0.9 | 46.1 ± 0.3 | **60.7 ± 1.2** | 19.4 ± 1.1 | **49.4 ± 2.1** | **42.7 ± 0.8** |
| ELICIT (diversity) | **42.6 ± 0.3** | **47.2 ± 0.1** | 54.8 ± 1.5 | **19.9 ± 0.8** | 47.1 ± 2.6 | 42.3 ± 0.9 |

## M  MULTI-LAYER INTERVENTION

While our primary analysis focuses on single-layer intervention, we also conduct preliminary experiments on multi-layer intervention, with the intervention strength $\alpha = 2$ distributed evenly across layers. We evaluated four settings: (1) the zero-shot baseline, (2) intervention on three consecutive layers (centered on the previously identified optimal layer), (3) intervention across all layers, and (4) our original single-layer implementation.

Results from Llama3-8B (Table 17) reveal an intriguing pattern: distributing intervention across multiple layers tends to yield better performance. This observation opens promising directions for future research into the mechanisms and benefits of multi-layer interventions.

Table 17: Comparison of multiple intervention layers on ELICIT. The experiments are conducted on Llama3-8B.

|  | nlu | reasoning | knowledge | math | safety | avg |
|---|---|---|---|---|---|---|
| zs | 32.4 | 31.8 | 42.8 | 15.4 | 36.6 | 31.8 |
| ELICIT (1 layer) | 38.3 | 46.9 | 60.7 | 20.6 | 51.1 | 43.5 |
| ELICIT (3 layers) | 38.2 | **47.1** | 61 | 21.6 | 51.6 | 43.9 |
| ELICIT (all layers) | **40.9** | 46.3 | **61.4** | **21.7** | **52.4** | **44.5** |

## N  ANALYSIS OF COMPUTATIONAL EFFICIENCY WITH RETRIEVAL MODULE

To demonstrate the effciency of ELICIT, We conducted a detailed analysis of ELICIT's computational efficiency using the Pythia-6.9B model, measuring the average processing time per sample across different pipeline stages. The results are shown in Table 18. Our quantitative results demonstrate that the integration of the retrieval module maintains the method's efficiency. Specifically, the retrieval module adds only 0.105 seconds of computational overhead per sample. The total inference time, including retrieval operations, remains efficient at 0.172 seconds per sample. ELICIT demonstrates superior efficiency compared to baseline approaches, processing samples 2-3 times faster than both 16-shot inference and BM25-based inference methods. These results validate that ELICIT's performance improvements do not come at the cost of computational efficiency, even with the addition of retrieval module.

Table 18: The running time of different stages per sample across different domains.

|  | zs inference time | ELCIT inference time | retrieve time | bm25 inference time | 16shot inference time |
|---|---|---|---|---|---|
| nlu | 0.063 | 0.064 | 0.097 | 0.302 | 0.181 |
| reasoning | 0.065 | 0.066 | 0.104 | 0.349 | 0.315 |
| knowledge | 0.066 | 0.069 | 0.108 | 0.517 | 0.371 |
| math | 0.065 | 0.067 | 0.111 | 0.351 | 0.352 |
| safety | 0.067 | 0.069 | 0.104 | 0.611 | 0.366 |
| avg | 0.065 | 0.067 | 0.105 | 0.426 | 0.317 |

