# OpenReview forum: "ELICIT: LLM Augmentation Via External In-context Capability"
_ICLR.cc/2025/Conference — ICLR 2025 Poster_

### Official Review · Reviewer_dv9u · 2024-10-30

**Soundness:** 3
**Presentation:** 3
**Contribution:** 3
**Rating:** 8
**Confidence:** 3

**Summary:**

This paper introduces ELICIT, a novel framework for adapting Large Language Models (LLMs) to diverse tasks, using task vectors and in-context learning (ICL). Inspired by the concept of modularization, ELICIT comprises two core components—Build Capability Library and Dynamic Capability Elicitation. By building a library of task vectors, each representing one in-context capability, ELICIT dynamically leverages this library to selectively retrieve and activate capabilities based on any given query, ensuring efficient and flexible elicitation of the model’s capabilities.

To build the capability library, task vectors are stored across layers for each task, along with prompts for future reuse, and the position of the optimal layer, which is determined based on the hold-out validation set and zero-shot inferences with task vector interventions to the model. Interventions within the model can be as direct replacement or a linear combination, with the latter shown to perform better. Later, task vectors are retrieved with the Dynamic Capability Elicitation module that employs a SimCSE RoBERTa model for relevant task vector selection and a threshold-based filtering approach based on AUC scores from the validation set.

ELICIT shows good performance across 20 ICL tasks and four models, outperforming other baselines across tasks, models and query formats, while also showing good generalization on unseen tasks.

**Strengths:**

- **S1:** Paper explores a novel and promising direction for in-context learning by leveraging task vectors with an external modular concept which is really interesting and aligns well with the recent work in the field of in-context learning and task-vectors.
- **S2:** Proposed method ELICIT demonstrates strong performance and generalization across diverse tasks and models.
- **S3:** This paper has a clear mathematical presentation with good explanations of ICL and task vectors.

**Weaknesses:**

- **W1:** The paper lacks specifics on dataset curation regarding the train-val-test splits and sizes, as well as the sample sizes used to calculate the performance. Can you please clarify how many examples were used for the task calculation?
- **W2:** ELICIT selects the optimal layer for task vectors for each given task and prompt, but assuming that the whole task encoding is stored just within a single layer may not be optimal. Have the authors tested multi-layer interventions, and if so, how did they compare? Moreover, could grouping and averaging task vectors per task simplify optimization steps and yield efficient task vector retrieval without extra steps?
- **W3:** The paper does not compare with previous task vector approaches mentioned in the related work (Hendel et al., Todd et al., Liu et al., Yan et al.), as well as with other parameter-efficient fine-tuning methods based on modularity (also mentioned in the related work like Hu et al.) with 16 shots. I believe such comparisons should be performed and included in the paper. Additionally, the paper does not mention task vector work for visual ICL and multi-modal ICL [1, 2] and including these would provide a comprehensive context and overview of the field.
- **W4:** Figure 3 illustrates the trade-off between stronger interventions and language modeling performance on WikiText, which is an expected observation since ICL and general language modeling operate with different circuits [3, 4, 5]. Having stronger interventions steers the activations further from the pretrained task, thus resulting in the worse performance, which is what previous works also showed. Authors do not analyze or explain this observation, but just comment how the strength of interventions affects the ICL and language modeling performances. I believe further discussion and explanation should be included.
- **W5:** The proposed method relies heavily on validation data to select optimal hyperparameters and determine the filtering threshold. And the similarity-based model for task vector retrieval is further trained. Does relying on the validation tuning affects the scalability and efficiency? Can you please explain how this similarity model was trained, and with what data?
- **W6:** Tables 1 and 2 contain additional bolded entries, and captions are not descriptive enough, missing information about the sample size for the BM25 and evaluation in general. Further, figure 6 does not have a clear explanation of components labeled a, b, and c, while also missing a description in general. Finally, there is a typo in the appendix in the title for the Similarity based retrieval methods.

[1] https://arxiv.org/abs/2404.05729

[2] https://arxiv.org/abs/2406.15334

[3] https://arxiv.org/abs/2205.05055

[4] https://arxiv.org/abs/2404.07129

[5] https://transformer-circuits.pub/2022/in-context-learning-and-induction-heads/index.html

**Questions:**

While the paper presents an interesting direction within ICL with modular task vectors and shows better performance than zero-shot and classical ICL, further improvement of clarity, related work comparison, and validation dependency could strengthen the paper even more. I would recommend acceptance if the authors address these points and the questions within the weakness section and the following one:
- Have you tried aggregating task vectors by task, and if so, what were the results?
- Have you considered using multiple layers to represent task vectors instead of relying on a single optimal layer? If yes, how did this affect performance?
- Can you please clarify if only the most similar task vector is used for intervention in the end? If so, does this mean many task vectors remain unused?
- Can you include comparisons to related task vector approaches from previous ICL research you mentioned in the related work? Can you also compare your method against PEFT modular methods with a few-shot regime?
- How well does your approach scale to larger LLMs?
- Can ELICIT be extended for multi-modal applications or other, more complex tasks?
- Does ELICIT support compositionality, such that combining different task vectors can represent new tasks?

---

> ### Author Response · Authors · 2024-11-23
> **Respnose to Reviewer dv9u (1/4)**
>
> Thank you for your time and insightful review. Following your insightful suggestions,
>
> 1.  We have conducted additional experiments, including implementing grouped ELICIT, testing with larger models, improving ELICIT by multi-layer intervention, and more complex task.
> 2. We have also updated our content to address your feedback regarding data curation, deeper interpretation and more complete related work.
>
> Hope these responses solve your concerns.
>
> **Weakness**
> > **W1:** The paper lacks specifics on dataset curation regarding the train-val-test splits and sizes, as well as the sample sizes used to calculate the performance. Can you please clarify how many examples were used for the task calculation?
> >
>
> Thank you for your suggestions! We followed a systematic approach for dataset splits, with two key principles:
>
> 1. Maintain a minimum test set size of 80 samples for evaluation.
> 2. Ensure train sets have at least 128 samples to enable ICL prompt construction
>
> **Our specific splitting strategies were as follows:**
>
> 1. Pre-existing Splits: For datasets like ARC-Challenge, Ethics, GLUE, MathQA, and OpenbookQA, we preserved their original train-val-test splits.
> 2. Train-Valid Only Datasets:
>     - For datasets with validation sets > 350 samples (e.g., BoolQ, Hellaswag): Split validation into new valid-test sets (7:3)
>     - For datasets with validation sets < 350 samples (e.g., CommonsenseQA): Split train set into train-test sets (7:3)
> 3. Test-Only Datasets:
>     - Small test sets (e.g., BBH with 250 samples):
>         - Train: 128 samples
>         - Test: 80 samples
>         - Remaining samples allocated to validation
>     - Large test sets (>1000 samples, e.g., MMLU-Pro-Math, BBQ, Crows Pairs):
>         - Split into train-valid-test (7:2:1)
> 4. Train-Only Datasets (e.g., SuperGLUE, DeepMind): Split into train-valid-test (7:2:1)
>
> We also added these details in Appendix I with red color. During library building we sampled from valid set to determine important hyparameters, and for evaluation we sampled 100 examples from the test set each time.
>
> > **W2:** ELICIT selects the optimal layer for task vectors for each given task and prompt, but assuming that the whole task encoding is stored just within a single layer may not be optimal. Have the authors tested **multi-layer interventions**, and if so, how did they compare? **Moreover, could grouping and averaging task vectors per task simplify optimization steps and yield efficient task vector retrieval without extra steps?**   (Q1+Q2)
> >
>
> See response to Q1 and Q2.
>
> > **W3:** The paper does not compare with previous task vector approaches mentioned in the related work (Hendel et al., Todd et al., Liu et al., Yan et al.), as well as with other parameter-efficient fine-tuning methods based on modularity (also mentioned in the **related work like Hu et al.**) with 16 shots. I believe such comparisons should be performed and included in the paper. Additionally, the paper does not mention task vector work for visual ICL and multi-modal ICL [1, 2] and including these would provide a comprehensive context and overview of the field.
> >
>
> Thank you for your advice. We excluded comparisons with other task vector approaches and PEFT methods since they are **orthogonal to our work** and serve as **alternative techniques** for creating capability library. We chose task vectors for their **simplicity and efficiency**, requiring only basic forward passes. In contrast:
>
> - [function vectors](https://arxiv.org/abs/2310.15213) require calculating significance metrics for all attention heads.
> - I[n-context vectors](https://arxiv.org/abs/2311.06668) need contrastive datasets to extract desired behavioral directions.
> - [state vectors](https://arxiv.org/abs/2404.11225) demand optimization.
> - PEFT methods with individual LoRA adaptors per task are more memory-intensive and need training.
>
> These methods involve more complex computations or modifications. Due to the time limit in the discussion period, we believe these methods can extend our work in the future.
>
> We also appreciate the suggestion to include visual and multi-modal ICL task vector research in our related work, which would provide broader context for the field. We added in Line 105.

---

> ### Author Response · Authors · 2024-11-23
> **Respnose to Reviewer dv9u(2/4)**
>
> > **W4:** Figure 3 illustrates the trade-off between stronger interventions and language modeling performance on WikiText, which is an expected observation since ICL and general language modeling operate with different circuits [3, 4, 5]. Having stronger interventions steers the activations further from the pretrained task, thus resulting in the worse performance, which is what previous works also showed. Authors do not analyze or explain this observation, but just comment how the strength of interventions affects the ICL and language modeling performances. I believe further discussion and explanation should be included.
> >
>
> Thank you for your valuable insights. This provides an interesting and plausible deeper understanding of the trade-off between stronger interventions and language modeling performance shown in Figure 3. We have added this possible interpretation in Appendix A.1.
>
> The results emerge from scenarios that differ from traditional ICL understanding. Our focus is on determining optimal intervention strengths for task vectors to elicit a model's inherent capabilities—an aspect that previous work has not comprehensively explored. This unexplored dimension significantly impacts our pipeline's effectiveness.
>
> > **W5:** **The proposed method relies heavily on validation data to select optimal hyperparameters and determine the filtering threshold**. And the similarity-based model for task vector retrieval is further trained. Does relying on the validation tuning affects the scalability and efficiency? Can you please explain how this similarity model was trained, and with what data?
> >
>
> That’s good question! We agree that our capability library construction takes time and relies on validation data, while
>
> - This construction is a one-time process and **doesn’t impact the efficiency during test time.**
> - Once constructed, task vectors can be **reused** during testing, improving overall efficiency.
> - We believe ELICIT **have scalability potential** when provided with a sufficient quantity and diversity of task vectors in the capability library. Because
>     - ELICIT can generalize to unseen tasks as shown in Table 3 in the paper.
>     - In Q7, we demonstrate that three types of task vectors could boost performance on an unseen task input.
>
> Instead of training a similarity-based model for retrieval, we directly compute similarities between:
>
> - Query embedding  $x \in \mathbb{R}^{L \times d}$
> - All task vectors $\{\theta_i\}_{i=1}^{k \times |\mathcal{T}|}$ in the library, where $\theta_i \in \mathbb{R}^{L \times d}$
>
> The similarity is computed using various metrics $f$, including:
>
> - Cosine similarity
> - Euclidean distance
> - t-SNE projection distance
>
> For each query , we obtain a similarity list of length $k \times |\mathcal{T}|$, where each element is $f(x,\theta_i)$. Prior to implementing these similarity metrics for retrieval, we evaluated their effectiveness through precision-recall AUC curves (shown in Figure 6). The analysis revealed that these similarity-based approaches demonstrate inadequate discrimination ability for identifying relevant task vectors from the library. This poor discriminative performance suggests that using these metrics for final retrieval and evaluation would not yield reliable or meaningful results.
>
> > **W6:** Tables 1 and 2 contain additional bolded entries, and captions are not descriptive enough, missing information about the sample size for the BM25 and evaluation in general. Further, figure 6 does not have a clear explanation of components labeled a, b, and c, while also missing a description in general. Finally, there is a typo in the appendix in the title for the Similarity based retrieval methods.
> >
>
> Thank you for your advice! We have made the modifications marked in red.

---

> ### Author Response · Authors · 2024-11-23
> **Respnose to Reviewer dv9u (3/4)**
>
> > **Q1**: Have you tried aggregating task vectors by task, and if so, what were the results?
> >
>
> Thank you for you advice!
>
> - We conducted preliminary experiments where we represent each task vector averaging by task with a single random ICL prompt. Then our grouped capability library consists of $(\hat{p}, \bar{\theta}, l^*)$), effectively reducing our task vectors from $|k \times \mathcal{T}|$ to $| \mathcal{T} |$.
>     - **Comparison Methods**: We compare
>         - zero-shot baseline
>         - *ELICIT (group+top-1)*: Use grouped capability library and choose top-1 task vector
>         - *ELICIT (group+top-2)*: Use grouped capability library and choose top-2 task vectors
>         - *ELICIT*: The original ELICIT implementation (without grouping)
>     - **Results**: The results on Pythia-6.9b showed in Table X12. **We find that our original implementation performs better overall.** Moreover, we observed that using grouped capability library harms performance in Knowledge and NLU tasks.
> - We did not average task vectors per task because **it would compromise our retrieval-based design.**
>     - Since our system handles queries from **unknown tasks**, we **require a retrieval module** to identify suitable task vectors from our capability library.
>     - Our retriever works by matching queries against individual ICL prompts to find the most similar examples. Averaging task vectors by task category would **break the one-to-one correspondence** between specific ICL prompts and their task vectors, undermining the retriever's ability to make precise matches.
>
> Table X12: The results of ELICIT using grouped capability library. The experiments are based on Pythia-6.9B.
>
> |  | nlu | reasoning | knowledge | math | safety | avg |
> | --- | --- | --- | --- | --- | --- | --- |
> | zs | 38.0 | 16.0  | 16.7  | 5.9  | 31.9  | 21.7  |
> | ELICIT (group+top-1) | *37.1* | 26.8  | 28.6  | **15.9**  | 49.0  | 31.5  |
> | ELICIT (group+top-2) | *37.1*  | 27.2  | *13.3*  | 14.4  | 49.1  | 28.2  |
> | ELICIT | **39.2**  | **27.4**  | **29.1**  | 15.3  | **49.6**  | **32.1**  |
>
> > **Q2:** Have you considered using multiple layers to represent task vectors instead of relying on a single optimal layer? If yes, how did this affect performance?
> >
> - **Multiple layer intervention shows promise**. It’s an interesting idea! We conducted preliminary experiments exploring this idea, where the intervention strength $\alpha=2$ was distributed equally across layers:
>     - **Comparison methods**: We conducted our experiments including the following settings:
>         - zero-shot baseline
>         - *ELICIT(1 layer)*: the original single-layer ELICIT implementation
>         - *ELICIT(3 layer)*: ELICIT intervention on 3 layers (centered on the optimal layer)
>         - *ELICIT(all layers)*: ELICIT intervention on all layers
>     - **Results**:  As shown in Table X4, the results from Llama3-8B demonstrate an interesting trend: increasing the number of layers involved in the intervention tends to improve overall performance. A deeper and more comprehensive investigation into this phenomenon remains an interesting direction for future research.
>
> We have added these results in Appendix M.
>
> Table X4: Comparison of multiple intervention layers on ELICIT. The experiments are conducted on Llama3-8B.
> |  | nlu | reasoning | knowledge | math | safety | avg |
> | --- | --- | --- | --- | --- | --- | --- |
> | zs | 32.4 | 31.8 | 42.8 | 15.4 | 36.6 | 31.8 |
> | ELICIT(1 layer) | 38.3 | 46.9 | 60.7 | 20.6 | 51.1 | 43.5 |
> | ELICIT(3 layers) | 38.2 | **47.1** | 61 | 21.6 | 51.6 | 43.9 |
> | ELICIT(all_layer) | **40.9** | 46.3 | **61.4** | **21.7** | **52.4** | **44.5** |
>
> > **Q3:** Can you please clarify if only the most similar task vector is used for intervention in the end? If so, does this mean many task vectors remain unused?
> >
>
> Thank you for your question!  We build a capability library containing various task vectors. Thus,
>
> - **For one query, only the similar task vectors are used during inference.** When processing an arbitrary query without explicit task information, we dynamically retrieve the most relevant task vector. The most similar task vectors are then used to elicit the model's inherent capability for that specific type of task.
> - **Different queries will trigger different task vectors from the library based on their specific requirements.** The capability library serve as a diverse pool of capabilities that can be activated for different types of queries. As shown in Figure 16 in Appendix J, we observe that 20 distinct types of task vectors are utilized after in-domain and out-of-domain (OOD) evaluations.
>
> We have added this result in Appendix J.

---

> ### Author Response · Authors · 2024-11-23
> **Respnose to Reviewer dv9u (4/4)**
>
> > **Q4:** Can you include comparisons to related task vector approaches from previous ICL research you mentioned in the related work? Can you also compare your method against PEFT modular methods with a few-shot regime?
> >
>
> see W3.
>
> > **Q5:** How well does your approach scale to larger LLMs?
> >
>
> Thank you for your advice! We conduct additional experiments on Pythia-12B and Llama3-70B.
>
> - **Results**: The results are shown in the Table X7. We can find ELICIT **is also effective for larger models.**
>
> We have added these results in Appendix K.
>
> Table X7: ELICIT performance on Pythia-12B and Llama3-70B. Pythia-12B on three seed while Llama 70B using 1 seed.
>
> |  |  | **Length** | **nlu** | **reasoning** | **knowledge** | **math** | **safety** | **avg** |
> | --- | --- | --- | --- | --- | --- | --- | --- | --- |
> | **Pythia-12B** | 16shots | 1598.2 ± 1.4 | 47.8 ± 1.2 | 15.8 ± 0.4 | 23.7 ± 0.3 | 14.3 ± 1.5 | 35.0 ± 1.9 | 27.3 ± 0.3 |
> |  | bm25 | 2184.2 ± 29.7 | 42.6 ± 1.4 | 20.3 ± 0.2 | 21.0 ± 0.4 | 14.0 ± 1.5 | 30.1 ± 0.8 | 25.6 ± 0.4 |
> |  | zs | 109.8 ± 1.5 | 34.7 ± 0.6 | 20.7 ± 0.2 | 18.1 ± 0.6 | 7.9 ± 1.7 | 34.6 ± 0.6 | 23.2 ± 0.2 |
> |  | ELICIT | 109.8 ± 1.5 | **38.5 ± 0.5** | **29.7 ± 0.7** | **29.8 ± 0.6** | **17.5 ± 2.1** | **46.8 ± 0.2** | **32.5 ± 0.5** |
> | **Llama3-70B** | bm25 | 1823.1 | 42.8 | 78.1 | 80.9 | 49.2 | 71.5 | 64.5 |
> |  | 16shot | 1411.5 | 40.1 | 76.3 | 83.3 | 51.4 | 70.9 | 64.4 |
> |  | zs | 101.1 | 50.9 | 66.8 | 59.7 | 37.6 | 44.2 | 51.8 |
> |  | ELICIT | 101.1  | **55.9** | **80.5** | **84.6** | **52.4** | **67.4** | **68.2** |
>
> > **Q6:** Can ELICIT be extended for multi-modal applications or other, more complex tasks?
> >
>
> Thanks for your advice. Multi-modality would be a good extension for ELICIT! As ELICIT has no constraint on the modality, we believe it can be extended to multi-modal tasks. We will investigate this direction as future work.
>
> Regarding for handling more complex tasks, we experiment on GSM8K, which contains  complex chain-of-thought generation and reasoning.
>
> - **Results:**  as shown in Table X13, ELICIT can handle GSM8K.
>
> Table X13: The results of ELICIT on GSM8K as in domain task based on Llama3-8B.
>
> |  | gsm8k |
> | --- | --- |
> | **zs** | 30.44 |
> | **ELICIT** | **32.44** |
> | **16shot** | 42.87 |
> | **16shot+ELCIT** | **43.22** |
>
> Our results can be found here and Appendix K in the paper.
>
> > **Q7:** Does ELICIT support compositionality, such that combining different task vectors can represent new tasks?
> >
>
> Indeed, while we did not explicitly design for compositionality initially, **our approach demonstrates emergent compositional properties.**
>
> - **The success of out-of-domain task handling using multiple task vectors already exhibits a form of compositionality**. To illustrate this, we present a case study using Pythia-6.9B selecting top-20 task vectors, where demonstrating combining different task vectors successfully represent an unseen DeepMind input.
>
> |  |  |
> | --- | --- |
> | input | The following are multiple choice questions (with answers) about algebraic word problems. Finish your answer with 'X' where X is the correct letter choice.\n\nQuestion: A sales staff is composed of a sales manager and two sales people, all of whom earn commission as a percentage of sales. Each sales person earns 5% commission on sales. In a given week, the sales staff earned a total of 2,500 in commissions on 5,000 worth of sales. What commission rate did the sales manager earn during that week?\nOptions:\nA. 25%\nB. 30%\nC. 35%\nD. 40%\nE. 45%\nAnswer: |
> | chosen task vectors | 10 MathQA task vectors + 2 CommonsenseQA task vectors + 8 BBH Boolean Expression task vectors |
> | Original Output | C |
> | ELICIT Output | **D (correct)** |
> |  |  |
>
> We believe ELICIT can be designed to achieve greater compositionality, which is an interesting exploration.

---

> ### Author Response · Authors · 2024-11-24
>
> We have submitted our revised manuscript along with additional experiments and responses to the questions. We kindly wanted to remind you, in case the notification was missed, and would greatly appreciate any updates on the responses. Thank you for your time!

---

> > ### Comment · Reviewer_dv9u · 2024-11-25
> >
> > Thanks for the detailed rebuttal, addressing my concerns and providing additional clarifications.
> >
> > I appreciate the additional experiments on scaling, multilayer interventions, and aggregation. I particularly find interesting the experiments regarding the multilayer interventions and composionality.
> >
> > After reading the other reviews and rebuttals, as well as checking the revised manuscript, with its improved visuals, explanations, and supplementary materials,  I believe the new version of the manuscript highlights the value of the proposed method and its potential impact in the community.
> >
> > I have decided to raise my score accordingly.

---

> > > ### Author Response · Authors · 2024-11-26
> > >
> > > Thank you for your valuable time and support in reviewing our manuscript. We are grateful for your positive evaluation and decision.

---

### Official Review · Reviewer_yoPv · 2024-11-02

**Soundness:** 3
**Presentation:** 3
**Contribution:** 2
**Rating:** 3
**Confidence:** 4

**Summary:**

The article introduces ELICIT, a novel framework to enhance the adaptive capabilities of large language models (LLMs) without the need for extensive fine-tuning or in-context learning demonstrations. ELICIT consists of two key modules: a capability library that stores task vectors representing various in-context learned capabilities, and a dynamic retrieval module that selectively activates these vectors based on input queries. Experimental results demonstrate the effectiveness of the model.

**Strengths:**

1. The new framework with the use of task vector demonstrates effectiveness in improving zero-shot performance.
2. The paper is generally well written and easy for readers to understand.

**Weaknesses:**

1. The novelty is limited in some aspects, including the use of task vector and the retrieval module.
2. Experiments on different models of different sizes should be conducted as the study would better demonstrate that this method is also effective for large models.
3. More comprehensive experiments on more datasets are expected, such as MMLU, GSM8K, HumanEval, etc.

**Questions:**

1. Why isn't the experiment conducted on instruction-tuned models but base models?
2. Did you compute the decrease in inference efficiency caused by the introduction of a new module?

---

> ### Author Response · Authors · 2024-11-23
> **Response to Reviewer yoPv (1/2)**
>
> Thank you for your time and reviews.
>
> **Weaknesses:**
>
> > W1. The novelty is limited in some aspects, including the use of task vector and the retrieval module.
> >
>
> We believe that our work makes concrete contributions to the community. To clarify,
>
> - Our work presents **a novel, modular framework** for enhancing the adaptive capabilities of LLMs on demand with minimal computational overhead, which approaches our **envisions** that ***LLMs should be capable of using its own best capabilities when solving problems.***
>     - The advantages of framework are validated though extensive experiments in our paper and acknowledged by all other Reviewers BquA, nS2i, dv9u.
> - **Different with previous work.** The line of work of task vectors focuses on single known task. In contrast, with our capability library, we now can **reuse** task vectors for arbitrary query without needing know what the task is.
> - This framework and vision represent **a significant step** forward in making language models more adaptable and efficient in real-world applications, acknowledged by Reviewers **dv9u and nS2i.**
>
> > W2. Experiments on different models of different sizes should be conducted as the study would better demonstrate that this method is also effective for **large models**.
> >
>
> Thank you for your advice! We conduct additional experiments on **larger models** Pythia-12B and Llama3-70B.
>
> - **Results**: The results are shown in the Table X7. We can find ELICIT is also effective for larger models.
>
> Table X7: ELICIT performance on Pythia-12B and Llama3-70B. Pythia-12B on three seed while Llama 70B using 1 seed.
>
> |  |  | **Length** | **nlu** | **reasoning** | **knowledge** | **math** | **safety** | **avg** |
> | --- | --- | --- | --- | --- | --- | --- | --- | --- |
> | **Pythia-12B** | 16shots | 1598.2 ± 1.4 | 47.8 ± 1.2 | 15.8 ± 0.4 | 23.7 ± 0.3 | 14.3 ± 1.5 | 35.0 ± 1.9 | 27.3 ± 0.3 |
> |  | bm25 | 2184.2 ± 29.7 | 42.6 ± 1.4 | 20.3 ± 0.2 | 21.0 ± 0.4 | 14.0 ± 1.5 | 30.1 ± 0.8 | 25.6 ± 0.4 |
> |  | zs | 109.8 ± 1.5 | 34.7 ± 0.6 | 20.7 ± 0.2 | 18.1 ± 0.6 | 7.9 ± 1.7 | 34.6 ± 0.6 | 23.2 ± 0.2 |
> |  | ELICIT | 109.8 ± 1.5 | **38.5 ± 0.5** | **29.7 ± 0.7** | **29.8 ± 0.6** | **17.5 ± 2.1** | **46.8 ± 0.2** | **32.5 ± 0.5** |
> | **Llama3-70B** | bm25 | 1823.1 | 42.8 | 78.1 | 80.9 | 49.2 | 71.5 | 64.5 |
> |  | 16shot | 1411.5 | 40.1 | 76.3 | 83.3 | 51.4 | 70.9 | 64.4 |
> |  | zs | 101.1 | 50.9 | 66.8 | 59.7 | 37.6 | 44.2 | 51.8 |
> |  | ELICIT | 101.1  | **55.9** | **80.5** | **84.6** | **52.4** | **67.4** | **68.2** |
>
> > W3. More comprehensive experiments on **more datasets a**re expected, such as MMLU, GSM8K, HumanEval, etc.
> >
>
> Thank you for your advice! We conducted additional experiments on more datasets:
>
> - We expand our existing capability library of Llama3-8B using GSM8K, which now consists of 21 tasks, and conduct experiments on GSM8K (in-domain) and a subset of MMLU (professional law as out-of-domain).
> - **Results**: We find that we could improve the performance on these two datasets while maintaining improvements on the other 25 tasks (20 in-domain tasks in Table 3 and 5 out-of-domain tasks in Table 4) both zero-shot and few-shot scenarios. It also shows the scalability of capability library.
>
> Table X8: The results of ELICIT on GSM8K and MMLU-Professional-Law on Llama3-8B. GSM8K is as in-domain task and MMLU-Profeesional-Law is out-of-domain.
> |  | gsm8k (in-domain) | mmlu-professional-law(ood) |
> | --- | --- | --- |
> | **zs** | 30.44 | 31.67 |
> | **ELICIT** | **32.44** | **41.11** |
> | **16shot** | 42.87 | 30.78 |
> | **16shot+ELCIT** | **43.22** | **41.89** |

---

> ### Author Response · Authors · 2024-11-23
> **Response to Reviewer yoPv (2/2)**
>
> **Questions:**
> > Q1. Why isn't the experiment conducted on instruction-tuned models but base models?
> >
> - We conducted a preliminary experiment on Llama3-8B-Instruct. The results shown in Table X9 demonstrates that **ELICIT can potentially work with instruction-tuned model**s.
>
>     Table X9: The preliminary experiment of ELICIT on Llama3-8B-Instruct.
>
>     |  | nlu | reasoning | knowledge | safety | avg |
>     | --- | --- | --- | --- | --- | --- |
>     | zs | 45.0  | 4.9  | 31.9  | 42.5  | 31.1  |
>     | Ours | **52.7**  | **36.2**  | **70.9**  | **49.0**  | **52.2**  |
> - We initially excluded instruction-tuned models due to **their sensitivity to prompts**, which would significantly increase the **complexity** of our experimental setup.
>     - **Evidence of sensitivity of Instruction-based model**: As demonstrated by prior work ([3,4]) and our additional case experiment with Llama3-8B-Instruct (Table X10), instruction-tuned models exhibit substantial variations in zero-shot performance based on prompt formatting or rephrasing.
>     - **This challenge is not related to our focus and will make experiments complex**. While our pipeline could be extended to instruction-tuned models, it would require additional considerations, particularly in the initial building stage where we need to identify effective task vectors for in-context learning (ICL). The challenge lies in determining optimal methods for concatenating examples and prompts for instruction-tuned models. By focusing on base models, we can conduct a more **straightforward analysis of our pipeline**'s effectiveness.
>
> Table X10: A case demonstrating Llama3-8B-Instruct's sensitivity to prompts.
>
> | input | output |
> | --- | --- |
> | prompt="<\|begin_of_text\|><\|start_header_id\|>system<\|end_header_id\|>\n\n**You are a pirate chatbot who always responds in pirate speak!**<\|eot_id\|><\|start_header_id\|>user<\|end_header_id\|>\n\n… input…<\|eot_id\|><\|start_header_id\|>assistant<\|end_header_id\|>\n\n" | Arrr, shiver me timbers! Yer |
> | "<\|begin_of_text\|><\|start_header_id\|>system<\|end_header_id\|>\n\n**You are a helpful assistant.**<\|eot_id\|><\|start_header_id\|>user<\|end_header_id\|>\n\n…input…<\|eot_id\|><\|start_header_id\|>assistant<\|end_header_id\|>\n\n" | B |
>
> [3] [Evaluating the zero-shot robustness of instruction-tuned language models](https://arxiv.org/abs/2306.11270)
>
> [4] [Roles of Scaling and Instruction Tuning in Language Perception: Model vs. Human Attention](https://aclanthology.org/2023.findings-emnlp.868/)
>
> > Q2. Did you compute the decrease in inference efficiency caused by the introduction of a new module?
> >
>
> In fact,
>
> - **quantitative results demonstrate the efficiency of our method even after introducing the retrieval module.** Using the Pythia-6.9B model, we measured the average processing time per sample across different pipeline stages (Table X11).
>     - Results show that our retrieval module introduces minimal computational overhead, **adding only 0.105 seconds** on average.
>     - The total ELICIT inference time, including retrieval, remains efficient at **0.172 seconds** per sample. Compared to baselines, ELICIT processes samples **2-3 times faster** than 16-shot inference or BM25 inference time.
>
> We have added add this analysis in Appendix N.
>
> Table X11: The running time of different stages per sample across different domains.
>
> |  | **zs inference time** | **ELCIT inference time** | **retrieve time** | **bm25 inference time** | **16shot inference time** |
> | --- | --- | --- | --- | --- | --- |
> | **nlu** | 0.063 | 0.064 | 0.097 | 0.302 | 0.181 |
> | **reasoning** | 0.065 | 0.066 | 0.104 | 0.349 | 0.315 |
> | **knowledge** | 0.066 | 0.069 | 0.108 | 0.517 | 0.371 |
> | **math** | 0.065 | 0.067 | 0.111 | 0.351 | 0.352 |
> | **safety** | 0.067 | 0.069 | 0.104 | 0.611 | 0.366 |
> | **avg** | 0.065 | 0.067 | 0.105 | 0.426 | 0.317 |
>
> Our work advances a novel vision for improving LLM: flexibly eliciting a model's relevant capabilities in response to arbitrary queries without requiring prior task identification. While previous task vector research has focused on known, single-task applications, we build a capability library that enables dynamic reuse of task vectors to explore the vision.
>
> Following your feedback, we have conducted comprehensive additional experiments with larger models and instruction-tuned models, and included detailed runtime analyses. And we updated the content accordingly. ***We hope these results resolve your concerns and will be really appreciated if you re-consider the score of our paper.***

---

> ### Author Response · Authors · 2024-11-24
>
> We have submitted our revised manuscript along with additional experiments and responses to the questions. We kindly wanted to remind you, in case the notification was missed, and would greatly appreciate any updates on the responses. Thank you for your time!

---

> ### Author Response · Authors · 2024-11-26
> **A Kind Reminder for Reviewer yoPv**
>
> Dear Reviewer yoPv,
>
> We would like to express our sincere gratitude for your thorough and insightful feedback regarding our manuscript. In response to the specific points you have raised, we have provided comprehensive explanations addressing each concern in detail. We summarize your questions and our key responses:
>
> - **[W1: Novelty of modules]:** We clarify our core contribution is the novel, modular framework for enhancing the adaptive capabilities of LLMs in low-computation, exploring our envisions that ***LLMs should be capable of using its own best capabilities when solving problems.*** Compared with previous work, ELICIT can reuse task vectors for arbitrary query without prior task information. This framework and vision represent **a significant step** forward in efficient LLM adaptation.
> - **[W2 & Q1: Effectiveness in larger models and instruction-tuned models]** We have conducted experiments to evaluate the effectiveness of ELICIT on more **diverse models** (Pythia-12B, Llama3-70B, Llama3-8B-Instruct). The results demonstrate that ELICIT can enhance the capabilities of these models. Additionally, we explain why we didn’t experiment on instruction-tuned models.
> - **[W3: Effectiveness on more datasets]** We have conducted experiments to assess the effectiveness on more datasets (GSM8K, MMLU-Professional-Law). The results demonstrate the **consistent improvements** of ELCIT on these datasets.
> - **[Q2: Inference Efficiency of Retrieval module]** We have analyzed the cost of retrieval module. The results demonstrate that the **minimal cost** (0.172 seconds) and together with inference, ELICIT is still **2-3 times** **faster** than baselines.
>
> We've incorporated your valuable suggestions into the revised manuscript and truly appreciate your thoughtful feedback.
>
> If you feel our responses have adequately addressed your concerns, we'd be grateful if you could consider revising the score. An improved score would be particularly important for our work at this stage.
>
> We're happy to address any additional questions or comments you might have. Your detailed review has helped us significantly improve our work, and we really appreciate the time you've invested. Looking forward to your feedback
>
> Best regards,
>
> Authors of Paper 10183

---

> ### Author Response · Authors · 2024-11-29
> **A kind reminder for Reviewer yoPv**
>
> Dear Reviewer yoPv,
>
> We'd like to send a gentle reminder that we have submitted our rebuttal addressing your comments. We sincerely appreciate your review and thoughtful feedback, which has helped us improve our manuscript.
>
> We are grateful that the other reviewers recognized the significance of our work, recommending acceptance with high scores of 8. We would appreciate the opportunity to discuss any remaining concerns and answer any further questions you may have.
>
> We thank you again for taking the time to review our work.
>
> Best regards,
>
> Authors of Paper 10183

---

> ### Author Response · Authors · 2024-12-03
>
> Dear Reviewer yoPv,
>
> We have revised the paper and added many additional results to address your comments. Since the rebuttal period is closing very soon, can you please check the response to see whether it mitigates your concerns? We would greatly appreciate that!
>
> Thank you,
>
> Authors of Paper 10183

---

### Official Review · Reviewer_nS2i · 2024-11-03

**Soundness:** 4
**Presentation:** 3
**Contribution:** 3
**Rating:** 8
**Confidence:** 3

**Summary:**

This paper proposes ELICIT. It's a framework aims at improving LLMs capabilities by introducing an external ICL capacity library. This library stores task vectors, which represent in-context learned abilities, enabling models to retrieve relevant skills dynamically without additional training tokens or fine-tuning. The approach allows LLMs to handle diverse tasks by selectively activating specific capabilities when needed, thus improving both versatility and computational efficiency.

**Strengths:**

1. The paper comes up with an interesting and intuitive solution to improve LLMs' abilities using the task vectors.
2. Extensive experiments over various models and tasks.
3. Exprimental results show great advantage of the method over others.
4. This novel plug-and-play framework could benefit other methods on the same task.

**Weaknesses:**

1. I would expect the proposed ELICIT method to be integrated into more existing strategies such as few-shot learning.
2. Have you tried using the capacity bank with other creation method other than ICL?

**Questions:**

See weakness above

---

> ### Author Response · Authors · 2024-11-23
> **Response Reviewer nS2i (1/1)**
>
> Thank you for reviewing and appreciating. We have added the suggested experiments and will update the paper accordingly. Please let us know if you have any additional questions or concerns.
>
> **Weaknesses:**
>
> > W1. I would expect the proposed ELICIT method to be integrated into more existing strategies such as few-shot learning.
> >
>
> Exactly, We are not sure about the specific few-shot learning approach you mentioned. We conducted additional experiments using ELICIT augmenting 16-shot in-context learning on Pythia-6.9B:
>
> - **Results:** The results in Table X6 demonstrate that ELICIT achieves performance improvements even in few-shot settings.
>
> If this wasn't what you meant, please let us know your specific thought about few-shot learning, and we'll be happy to verify it accordingly.
>
> Table X6: ELICIT as an plug-and-play performance booster: Performance Gains When Combined with 16-shot ICL on In-Domain Tasks. The experiments are conducted on Pythia-6.9B.
>
> |  | **Length** | **nlu** | **reasoning** | **knowledge** | **math** | **safety** | **avg** |
> | --- | --- | --- | --- | --- | --- | --- | --- |
> | **16shot** | 1595.6 | **46.9** | 23.8 | 22.2 | 10.9 | 34.3 | 27.7 |
> | **16shot + ELICIT** | 1595.6 | 42.1 | **26.3** | **25.3** | **13.8** | **39.4** | **29.4** |
>
> > W2. Have you tried using the capacity bank with other creation method other than ICL?
> >
>
> We agree there exists more creation methods building capability library beyond ICL, such as prompt optimization. We haven't explored these approaches yet because our initial focus was on using ICL to investigate the possibility of enhancing LLMs' adaptive capabilities for arbitrary queries with minimal computational overhead. While we acknowledge that extending this work to other methods could be valuable, we consider it as an interesting direction for future research.

---

> > ### Comment · Reviewer_nS2i · 2024-11-27
> >
> > Thanks for the response, authors have partially addressed my concerns and left the rest for future explorations. I'll keep my score.

---

> > > ### Author Response · Authors · 2024-11-27
> > >
> > > Thank you for taking the time to provide such constructive feedback and for recommending acceptance. Your insights and support mean a great deal to us.

---

### Official Review · Reviewer_BquA · 2024-11-04

**Soundness:** 3
**Presentation:** 3
**Contribution:** 3
**Rating:** 8
**Confidence:** 3

**Summary:**

This paper proposes ELICIT, a framework that stores the task vectors corresponding to different in-context-learning (ICL) prompts and dynamically augments the given question with retrieved task vectors to provide ICL abilities without explicitly forwarding the long ICL prompts. ELICIT shows comparable or better performance than ICL baselines on diverse tasks while being significantly more efficient.

**Strengths:**

1. The writing is clear and easy to follow. The motivation and design of each component is straightforward to understand.
2. Dynamically augmenting task vectors is significantly more efficient than in-context learning while showing competitive or even better performance.
3. The proposed approach can be applied to existing LLMs in a plug-and-play manner, making ELICIT easy to deploy.

**Weaknesses:**

1. Some details regarding the experiment setup need to be included. For example, the paper does not describe how the ICL prompts $p_i^{t}$ are chosen.

**Questions:**

1. How are the ICL prompt $p_i^{t}$ chosen? (e.g., Are they a group of randomly selected examples?) If there is a technique to maximize the diversity of the prompts in a given library, would it also boost the model performance?
2. Figure 5 shows that ELICIT boosts performance on relevant tasks while minimally compromising performance on non-related tasks. Is this because task vectors are not applied for unrelated tasks, or does the model perform well even when unrelated task vectors are given?
3. The current framework augments (at most) a single task vector at a single layer. Could this be extended to multiple vectors or layers to improve performance? (e.g., in cases where a given query is highly relevant to two different task vectors)
4. Tables 2 and 3 show that ELICIT performs significantly better than the ICL baselines for Pythia and Mamba. What would be the reason for this? For example, can ELICIT be more beneficial for models with weaker capabilities? Or could this be related to a specific training recipe, as Pythia and Mamba are trained under the same setup?

---

> ### Author Response · Authors · 2024-11-23
> **Response to Reviewer BquA (1/3)**
>
> Thank you for your thoughtful review and suggestions. We have incorporated additional experiments based on your feedback and revised the manuscript accordingly. Please let us know if you have any further questions or suggestions.
>
> **Weaknesses:**
>
> W1. Some details regarding the experiment setup need to be included. For example, the paper does not describe how the ICL prompts pit are chosen.
>
> See [Q1](#q1).
>
> **Questions:**
>
>
> > Q1: How are the ICL prompt pit chosen? (e.g., Are they a group of randomly selected examples?) If there is a technique to maximize the diversity of the prompts in a given library, would it also boost the model performance?
> >
>
> Thanks for your advice! We have clarified in Line 239 that the demonstrations in the ICL prompts are randomly selected.
>
> Maximizing the diversity of the prompts in a given library is a good idea! We conduct an additional experiment with embedding diversity, comparing random demonstration selection with diversity-optimized prompts as described in this [paper](https://arxiv.org/pdf/2209.01975).
>
> - **Comparison Methods**: we compare
>     - *Zero-shot*: zero-shot baseline
>     - *ELICIT*: original ELICIT implementation with randomly selected ICL demonstrations
>     - *ELICIT (diversity)*: modified ELICIT using the new capability library with diversity-optimized demonstrations.
> - **Results:** As shown in Table X1, **the diversity-optimized prompts work well in some cases but not in others.** Compared to the original ELICIT, while performance improved in reasoning (+1.1%), math (+0.5%) and NLU tasks (+4.5%), there was a decline in Knowledge (-5.9%) and Safety (-2.3%) ability.
>
> This result suggests the **potential** for future work to **improve our pipeline** by enhancing the quality of task vectors through better demonstration selection methods.
>
> Table X1: The comparison of ELICIT using different capability library based on different ICL prompts. The experiments are conducted on Llama3-8B.
>
> |  | **NLU** | **Reasoning** | **Knowledge** | **Math** | **Safety** | **Avg.** |
> | --- | --- | --- | --- | --- | --- | --- |
> | Zero-shot | 32.2 ± 1.2 | 32.9 ± 0.2 | 42.5 ± 1.2 | 14.0 ± 1.0 | 35.5 ± 1.2 | 31.4 ± 0.7 |
> | ELICIT | 38.1 ± 0.9 | 46.1 ± 0.3 | **60.7 ± 1.2** | 19.4 ± 1.1 | **49.4 ± 2.1** | **42.7 ± 0.8** |
> | ELICIT (diversity) | **42.6 ± 0.3** | **47.2 ± 0.1** | 54.8 ± 1.5 | **19.9 ± 0.8** | 47.1 ± 2.6 | 42.3 ± 0.9 |
>
> We have added these results in Appendix L.

---

> ### Author Response · Authors · 2024-11-23
> **Response to Reviewer BquA (2/3)**
>
> > Q2. Figure 5 shows that ELICIT boosts performance on relevant tasks while minimally compromising performance on non-related tasks. Is this because task vectors are not applied for unrelated tasks, or does the model perform well even when unrelated task vectors are given?
> >
>
> **This phenomenon occurs because task vectors are not applied for unrelated queries. And when we forcibly applying task vectors to all queries can actually harms model performance.** We conducted additional experiments:
>
> - We calculate the number of chosen states per domain for each sample when the library only contains math-related task vectors on Mistral, the results are presented in Table X2.
>     - **Results**: The result shows that math-related tasks consistently employ a high number of chosen states (9.8 ± 0.1), while other domains show minimal state selection (close to 0.0). This result demonstrates that the behavior observed in Figure 5 arises from ELICIT's ability to dynamically retrieve and reuse task vectors from the capability library, enabling **selective activation** of relevant capabilities.
>     - **Case Study on Unrelated domains**: We observed minor improvements in reasoning tasks, exemplified by this ARC Challenge case. It demonstrates our pipeline's ability to selectively activate relevant capabilities based **solely on query and handle unseen tasks flexibly, without requiring explicit task information.**
>
> |  |  |
> | --- | --- |
> | input | Below are multiple-choice science questions. Answer with 'X', X being the correct option.\n\nQuestion: An unbalanced equation for the reaction of methane gas (CH_{4}) with oxygen is shown below. CH_{4} + \\Box O_{2} -> 2CO_{2} + 4H_{2}O How many molecules of oxygen gas (O_{2}) are needed to properly balance this equation?\nOptions:\nA. 1\nB. 2\nC. 3\nD. 4\nAnswer: |
> | chosen task vectors | 10 task vectors from MathQA |
> | Original Output | B |
> | ELICIT Output | **D (correct)** |
> |  |  |
> - **Forcibly applying the top task vectors for each query can harm performance.** We conduct experiments on Mistral (Table X3) showed that this approach led to significant declines in NLU and knowledge performance.
>
> These experimental results demonstrate that ELICIT's performance improvement stems from its selective activation mechanism and the importance of selectively using only task-relevant vectors to dynamically activate capabilities.
>
> We have added these results in Appendix J and add description in main content Line 456 to clarify that.
>
> Table X2: The average number of chosen numbers per domain per sample. The statistics come from Mistral when the capability library only contains math-related task vectors.
>
> |  | in-domain |  |  |  |  |
> | --- | --- | --- | --- | --- | --- |
> |  | **NLU** | **Reasoning** | **Knowledge** | **Math** | **Safety** |
> | chosen nums | 0.0 ± 0.0 | 0.1 ± 0.0 | 0.0 ± 0.0 | **9.8 ± 0.1** | 0.0 ± 0.0 |
> |  | **Out-of-domain** |  |  |  |  |
> |  | **GLUE COLA** | **BBQ Religion** | **Deepmind** | **MMLU-Psychology** | **BBH-five-objects** |
> | chosen nums | 0.0 ± 0.0 | 0.0 ± 0.0 | **9.9 ± 0.1** | 0.0 ± 0.0 | 0.0 ± 0.0 |
>
> Table X3: The results of forcibly applying the top task vectors for each query. The experiments were conducted on Mistral. Domains with degraded performance are marked in *italics*.
> |  | nlu | reasoning | knowledge | math | safety |
> | --- | --- | --- | --- | --- | --- |
> | zs | 28.8 | 27.4 | 58.8 | 4.0 | 42.2 |
> | Ours | *15.7* | 31.4 | *47.8* | 18.3 | 53.1 |

---

> ### Author Response · Authors · 2024-11-23
> **Response to Reviewer BquA (3/3)**
>
> > Q3. The current framework augments (at most) a single task vector at a single layer. Could this be extended to multiple vectors or layers to improve performance? (e.g., in cases where a given query is highly relevant to two different task vectors)
> >
>
> Thank you for advice! We consider for these two potential improvements as following:
>
> - **Appropriate numbers of chosen task vectors are important**. As mentioned in Line 340, we selected n=10 task vectors from the library during evaluation, enabling selection from different tasks. A simple ablation study in Section 5.2 demonstrates that n=10 yields optimal performance compared to other values.
> - **Multiple layer intervention shows promise**. We conducted preliminary experiments to explore this idea, where the intervention strength $\alpha=2$ was distributed equally across layers:
>     - **Comparison methods**: We conducted our experiments including the following settings:
>         - *zs*: zero-shot baseline
>         - *Ours (1 layer)*: the original single-layer implementation
>         - *Ours (3 layers)*:  intervention on 3 layers (centered on the optimal layer)
>         - *Ours (all_layer)*: intervention on all layers
>     - **Results**: As shown in Table X4, the results from Llama3-8B demonstrate an interesting trend: increasing the number of layers involved in the intervention tends to improve overall performance. A deeper and more comprehensive investigation into this phenomenon remains an interesting direction for future research.
>
> We have added these results in Appendix M.
>
> Table X4: Comparison of multiple intervention layers on ELICIT. The experiments are conducted on Llama3-8B.
> |  | nlu | reasoning | knowledge | math | safety | avg |
> | --- | --- | --- | --- | --- | --- | --- |
> | zs | 32.4 | 31.8 | 42.8 | 15.4 | 36.6 | 31.8 |
> | Ours (1 layer) | 38.3 | 46.9 | 60.7 | 20.6 | 51.1 | 43.5 |
> | Ours (3 layers) | 38.2 | **47.1** | 61 | 21.6 | 51.6 | 43.9 |
> | Ours (all_layer) | **40.9** | 46.3 | **61.4** | **21.7** | **52.4** | **44.5** |
>
> > Q4. Tables 2 and 3 show that ELICIT performs significantly better than the ICL baselines for Pythia and Mamba. What would be the reason for this? For example, can ELICIT be more beneficial for models with weaker capabilities? Or could this be related to a specific training recipe, as Pythia and Mamba are trained under the same setup?
> >
>
> In principle, this observation should **relate to the model’s weak capability of incorporating contextual information.** To further investigate the reason, we conduct scaling experiments on larger versions of Pythia, ranging from 2.8b to 12b.
>
> - **Results**: The results are shown in Table X5. We find that even though the ICL performance increases with the increases of sizes, our method still outperforms the ICL baseline for all sizes. We cannot draw a conclusion now based on our results, and these results show that at least the reason **cannot be simply explained by the scaling of model sizes**. As various version of Pythia intuitively would follow the same training recipe, **we now incline to the hypothesis that a certain setup leads to weak contextual capability.**
>
> This is an intriguing question that requires deeper investigation and better understanding. We believe this is a valuable direction for future research.
>
> Table X5: The results of ELICIT on Pythia series ranging from 2.8b to 12b.
> |  |  | Length | nlu | reasoning | knowledge | math | safety | avg |
> | --- | --- | --- | --- | --- | --- | --- | --- | --- |
> | Pythia-2.8b | 16-shot | 1597.6 ± 2.4 | 48.1 ± 0.4 | 22.2 ± 0.8 | 12.5 ± 0.7 | 10.3 ± 0.9 | 28.2 ± 0.9 | 24.3 ± 0.4 |
> |  | ELICIT | 109.8 ± 1.5 | **60.1 ± 0.1** | **25.7 ± 0.9** | **20.9 ± 1.2** | **14.4 ± 1.3** | **40.9 ± 2.5** | **32.4 ± 0.4** |
> | Pythia-6.9b | 16shots | 1598.4 ± 2.0 | 27.3 ± 0.3 | 27.3 ± 0.3 | 27.3 ± 0.3 | **27.3 ± 0.3** | 27.3 ± 0.3 | 27.3 ± 0.3 |
> |  | ELICIT | 109.8 ± 1.5 | **38.7 ± 1.4** | **28.1 ± 0.5** | **27.9 ± 1.0** | 18.2 ± 2.6 | **47.8 ± 2.0** | **32.2 ± 0.7** |
> | Pythia-12b | 16shots | 1598.2 ± 1.4 | **47.8 ± 1.2** | 15.8 ± 0.4 | 23.7 ± 0.3 | 14.3 ± 1.5 | 35.0 ± 1.9 | 27.3 ± 0.3 |
> |  | ELICIT | 109.8 ± 1.5 | 38.5 ± 0.5 | **29.7 ± 0.7** | **29.8 ± 0.6** | **17.5 ± 2.1** | **46.8 ± 0.2** | **32.5 ± 0.5** |

---

> ### Comment · Reviewer_BquA · 2024-11-26
>
> Thank you for the clarifications and for your efforts on extensive additional experiments. My concerns are resolved.

---

> > ### Author Response · Authors · 2024-11-26
> >
> > Thank you for your valuable time and support in reviewing our manuscript. We are grateful for your positive evaluation and decision.

---

### Author Response · Authors · 2024-11-23
**General Response**

We sincerely appreciate all the reviewers for their time and insightful reviews. We are encouraged that the reviewers acknowledged our presentation is clear and easy to follow (Reviewers BquA, yoPv, dv9u), the motivation is straightforward (Reviewer BquA), the improvements on zero-shot performance are significant and consistent (Reviewers nS2i, yoPv, dv9u), the proposed method is efficient and flexible (Reviewers BquA, nS2i), and the experiments are extensive (Reviewer nS2i).

We are particularly grateful that reviewers recognized our core contributions: advancing novel approaches to improve LLM capabilities (Reviewer nS2i) and exploring a promising new direction in the field (Reviewer dv9u).

### Summary of Contribution and Novelty

We believe that our work makes concrete contributions to the community.

To clarify, our work presents a novel, modular framework ELICIT for enhancing the adaptive capabilities of LLMs on demand with minimal computational overhead, which approaches our envisions that ***LLMs should be capable of using its own best capabilities when solving problems during test time,*** aligning with the recent trend of test-time scaling for language models [1, 2].

With our capability library, our framework can reuse task vectors for arbitrary query **without requiring prior task identification**, which is different from previous work. ***This framework and vision represent a significant step*** forward in making language models more adaptable and efficient in real-world applications.

[1] [Scaling LLM Test-Time Compute Optimally can be More Effective than Scaling Model Parameters](https://arxiv.org/abs/2408.03314)

[2] [O1 Replication Journey: A Strategic Progress Report – Part 1](https://arxiv.org/abs/2410.18982)

### Summary of Additional Results During Discussion Period

We conducted extensive supplementary experiments covering:

- **Model scaling and Diversity (Appendix K)**
    - An additional base model (Pythia-6.9B)
    - Larger models (Pythia-12b and Llama3-70B)
    - An instruction-tuned model (Llama3-8B-Instruct)
- **More tasks (Appendix K)**
    - A complex task (GSM8K)
    - A Specialized Out-Of-Domian task (MMLU-Professional Law)
- **Deeper Analysis of framework**
    - Analysis of task vectors usage behavior. (Appendix J)
    - Forcibly applying the top task vectors for each query. (Appendix J)
    - Efficiency analysis of adding a retrieval module. (Appendix N)
- **Integration with one more existing Technique**
    - 16-shot ICL + ELICIT.
- **The potential way to improve framework**
    - Results on Diversity-optimized capability library. (Appendix L)
    - Multi-layer intervention. (Appendix M)
    - Results on task-grouped capability library.

We sincerely thank all reviewers for their insightful suggestions. These results have been added to the appendix accordingly, and we have addressed the writing-related feedback in the paper (highlighted in red).

---

### Meta-Review · Area_Chair_wSmS · 2024-12-18

**Metareview:**

This paper proposes a new approach that improves LLMs capabilities by introducing an external ICL capacity library.

Three reviewers support the contributions of this paper with clear cceptance scores while one reviewer gave clear rejection.

AC carefully read the paper, reviewers' comments and author feedback.

While R-yopv pointed out major concerns such as limited novelty and insufficient experiments, the authors provide the additional experimental results on those issues. Although yopv does not show his/her feedback, AC thinks that the authors successfully addressed the concerns.

So, AC recommends accepting this paper.

**Additional Comments On Reviewer Discussion:**

The initial scores were 8, 8, 3, and 5.

Negative reviwers raised some major concerns related to experiments with restricted novelty.

During the rebuttal period, the authors presented extensive addtional experimental results, so addressed the concerns of nS2i.
Thus, nS2i increased his score from 5 to 8.

yoPv did not show any response on the authors' comments.

The final scores are 8, 8, 3, and 8.

---

### Decision · Program_Chairs · 2025-01-22

Accept (Poster)